# Understanding the Role of Training Data in Test-Time Scaling

**Adel Javanmard[1,3], Baharan Mirzasoleiman[2,3], Vahab Mirrokni[3]**
[1]University of Southern California, [2]University of California Los Angeles, [3]Google Research
`ajavanma@usc.edu, baharan@cs.ucla.edu, mirrokni@google.com`

## Abstract

Test-time scaling improves the reasoning capabilities of large language models (LLMs) by allocating extra compute to generate longer Chains-of-Thoughts (CoTs). This enables models to tackle more complex problem by breaking them down into additional steps, backtracking, and correcting mistakes. Despite its strong performance–demonstrated by OpenAI's o1 and DeepSeek R1, the conditions in the training data under which long CoTs emerge, and when such long CoTs improve the performance, remain unclear. In this paper, we study the performance of test-time scaling for transformers trained on an in-context weight prediction task for linear regression. Our analysis provides a theoretical explanation for several intriguing observations: First, at any fixed test error, increasing test-time compute allows us to reduce the number of in-context examples (context length) in training prompts. Second, if the skills required to solve a downstream task are not sufficiently present in the training data, increasing test-time compute can harm performance. Finally, we characterize task hardness via the smallest eigenvalue of its feature covariance matrix and show that training on a diverse, relevant, and hard set of tasks results in best performance for test-time scaling. We confirm our findings with experiments on large, nonlinear transformer architectures.

## 1 Introduction

Scaling test-time compute enhances inference in large language models (LLMs), by enabling reasoning with long chains-of-thought (CoTs). This allows models to generate more intermediate reasoning steps for complex problems, evaluate multiple options, and backtrack to find more accurate answers, all without changing the model's parameters. There has been a recent body of work on this idea (Snell et al., 2024; Welleck et al., 2024; Muennighoff et al., 2025; Yeo et al., 2025), with OpenAI's o1 (OpenAI, 2024) and DeepSeek R1 (Guo et al., 2025) demonstrating strong reasoning performance with consistent gains from scaling test-time compute. However, our understanding of the training data properties that support test-time scaling remains limited.

Training on diverse and difficult data has shown to be beneficial to enable test-time scaling on complex reasoning tasks, such as mathematical competitions (Muennighoff et al., 2025), medical reasoning (Huang et al., 2025b), and code (Yu et al., 2025). Difficult examples are often identified as those that cannot be answered by the model being trained or other more powerful proxy models. However, the precise notion of difficulty and the relation between the amount of compute at training and test time remains unclear. In particular,

   $(i)$ Does increasing the test-time compute always improve the downstream reasoning performance?

  $(ii)$ Can increasing the test-time compute lower the requirement on training-time compute?

 $(iii)$ What are difficult training examples and why are they beneficial for test-time scaling? Addressing this question requires a rigorous understanding of the effect of training data and its properties on the performance of test-time scaling.

In this paper, we theoretically study the performance of test-time scaling for transformers trained on an in-context weight prediction task for linear regression, where the goal is to predict the linear weight

vector from the sequence of input prompts. This framework has been used previously for analyzing the mechanism underlying training CoT (Huang et al., 2025a). During training, the model performs direct in-context-learning and outputs its prediction of the weight vector. At test time, the transformer performs CoT and generates multiple intermediate steps before arriving at its final prediction of the weight vector. Our analysis yields several intriguing findings: First, fixing the test error, by increasing the test-time compute we can decrease the number of in-context examples (context length) in training prompts. Second, if the skills needed to solve the downstream task (corresponding to directions in the data covariance matrix) are not sufficiently represented in the training data, increasing test-time compute can harm performance, effectively causing the model to *overthink*. Finally, we characterize hardness of a task based on the smallest eigenvalue of its feature covariance matrix and show that training on a diverse, relevant and hard set of tasks during training yields the best performance for test-time scaling.

Our **main contributions** and the organization of the paper are discussed below:

$(a)$ In Section 3, We study in-context learning in transformers with a single linear self-attention (LSA) layer trained via gradient descent. Despite the problem's non-convexity, we show that gradient descent, when initialized randomly but suitably, converges to a global minimum, which we explicitly characterize. Our analysis allows for general feature covariance. During training, the model engages in direct in-context learning, but at test time we employ chain-of-thought (CoT) prompting to let the model generate intermediate reasoning steps before producing its final output. We demonstrate that, with CoT prompting at test time, the transformer effectively implements a multi-step (pseudo-) Newton's method for loss optimization. Notably, this part of our contribution extends the results of Zhang et al. (2024) by incorporating CoT dynamics at test time, and of Huang et al. (2025a) by accommodating general feature covariance.

$(b)$ By analyzing the expected estimation error of in-context weights for test prompts, in Section 3.4 we introduce a measure of task hardness defined by the ratio of the smallest eigenvalue of the feature covariance matrix to its trace. We interpret the eigenvectors as representing different skills relevant to the task, with the corresponding eigenvalues indicating the strength of those skills. Under this interpretation, hard tasks are characterized by a long-tailed spectrum of skills, while easy tasks correspond to having only a few well-balanced skills.

This analysis leads to two key consequences: (1) For a fixed test error, *increasing test-time compute allows us to reduce the required number of in-context examples* (i.e., the context length) in training prompts. (2) We derive **test-time scaling laws** for our ICL setting, capturing how test error depends on test-time compute and highlighting the role of factors such as context length, feature dimension, and task covariance structure in shaping the overall trend.

$(c)$ In Section 5.1, we study a setting with $T$ tasks, where each task is specified by its feature covariance matrix (interpreted, as discussed in part (b), as the set of skills required for the task together with their relative strengths). We extend the analysis of Section 3 to this multi-task setting and characterize the estimation error of the final CoT output. Based on this characterization, we formulate a quadratic optimization problem to determine the optimal task selection probabilities, demonstrating that *training on a diverse, relevant, and sufficiently hard set of tasks yields the best performance under test-time scaling.* We validate our theoretical results with experiments on both Linear Self-Attention (LSA) models and the more complex nonlinear transformer architecture GPT-2.

## 2 RELATED WORK

Recent work has highlighted several phenomena relevant to our study. First, it has been observed that simply increasing test-time compute and reasoning depth can, counterintuitively, harm performance, a phenomenon termed *overthinking*. The empirical study of Su et al. (2025) suggests that LLMs tend to overthink simple problems by generating unnecessarily long outputs, and underthink harder ones, by providing shallow or incomplete reasoning that overlooks critical steps. In Wang et al. (2025), it is argued that exploring more reasoning branches may degrade system efficiency as many branches may be trapped in overthinking. Second, Recent work has explored the test-time scaling paradigm (Snell et al., 2024; Welleck et al., 2024), with OpenAI's o1 (OpenAI, 2024) and DeepSeek R1 (Guo et al., 2025) demonstrating strong performance through reinforcement learning on millions of samples and multiple training stages. Muennighoff et al. (2025) proposes a simple framework,

which involves training on only 1,000 samples with next-token prediction and controlling thinking duration via a simple test-time technique, and show that it achieves test-time scaling and strong reasoning performance. Finally, prior studies on data mixtures emphasize the importance of balancing training corpora with sufficient coverage of topics matched to downstream tasks, as imbalanced data composition can impair generalization (Xie et al., 2023; Nguyen et al., 2024). However, prior work has been largely empirical, whereas we develop a theoretical framework that rigorously analyzes test-time scaling and Chain-of-Thought effectiveness, overthinking, and principled strategies for task selection during training.

## 3 IN-CONTEXT LEARNING

In an in-context learning (ICL) scenario, a model is presented with instances of prompts of the form $P_\tau = (x_1, h_\tau(x_1), \ldots, x_n, h_\tau(x_n))$, with $x_i$ drawn i.i.d from a distribution $\mathcal{D}_x$, and $h_\tau$ sampled independently from a distribution over functions in a given function class. The goal of in-context learning is to train a model so that when given a test prompt $P_{\tau'} = (x_1, h_{\tau'}(x_1), \ldots, x_m, h_{\tau'}(x_m), x_{m+1})$ with an independently sampled $h_{\tau'}$, it is able to make a prediction on $x_{m+1}$ that is close to $h_{\tau'}(x_{m+1})$. Therefore, a key distinction from traditional supervised learning is that in ICL, each prompt has its own distribution. For example, in linear regression, $h_\tau(x) = \langle w_\tau, x \rangle$, where each prompt has its own ground truth $w_\tau$. Thus, in ICL the model must generalize not just across data points but across distributions, and be able to infer the correct predictive rule on the fly for each new prompt without modifying its parameters.

### 3.1 IN-CONTEXT WEIGHT PREDICTION AND LINEAR SELF-ATTENTION

We focus on ICL for linear regression task, where each prompt $P_\tau = (x_{\tau,1}, y_{\tau,1}, \ldots, x_{\tau,n}, y_{\tau,n})$ with $y_\tau = \langle w_\tau, x_{\tau,i} \rangle$, where $x_{\tau,i} \sim \mathsf{N}(0, \Lambda)$, $w_\tau \sim \mathsf{N}(0, I_d)$. Most previous works on this setting focus on prediction without directly estimating the weight vector of the test prompt (Ahn et al., 2023a; Zhang et al., 2024; Mahankali et al., 2023). Here, we take a similar approach to Huang et al. (2025a) and consider in-context weight prediction where we require the model to directly estimate the wight vector of test prompts. To this end, we adopt the embedding used by Bai et al. (2023); Huang et al. (2025a) which includes the weight-estimation:

$$E_\tau = \begin{bmatrix} x_{\tau,1} & \cdots & x_{\tau,n} & 0 \\ y_{\tau,1} & \cdots & y_{\tau,n} & 0 \\ 0 & \cdots & 0 & \hat{w}_0 \\ 0 & \cdots & 0 & 1 \end{bmatrix} := \begin{bmatrix} X_\tau & 0 \\ y_\tau & 0 \\ 0_{d\times n} & \hat{w}_0 \\ 0_{1\times n} & 1 \end{bmatrix} \tag{3.1}$$

where $\hat{w}_0 \in \mathbb{R}^d$ is an initialization for the weight estimate.

We next proceed by describing the transformer architecture. We consider a one layer self-attention with residual connection. Let $E$ be an embedding formed from the prompt. A self-attention module takes as input an embedding matrix and outputs a matrix of the same size,

$$f_{\text{Attn}}(E; W_K, W_Q, W_V, W_P) = E + W_P W_V E \cdot \psi\left(\frac{(W_K E)^\top W_Q E}{\rho}\right)$$

where $\psi$ is an activation (e.g. softmax) that is applied column-wise. Following Gatmiry et al. (2024); Huang et al. (2025a); Zhang et al. (2024); Ahn et al. (2023b), we consider Linear-Self-Attention (LSA) where the activation $\psi$ is the identity mapping. By defining $W := W_K^\top W_Q$, $V = W_P W_V$ and $\theta = (W, V)$ we arrive at

$$f_{\text{LSA}}(E; \theta) = E + VE \cdot \frac{E^\top W E}{\rho}. \tag{3.2}$$

The estimation of the transformer for $w_\tau$ is given by the last token of the output sequence, namely $\hat{w}_\tau = f_{\text{LSA}}(E_\tau; \theta)_{[d+2:2d+1, -1]}$, which is obtained by restricting the last column of $f_{\text{LSA}}(E_\tau; \theta)$ to entries $[d+2 : 2d+1]$. We assume $\hat{w}_0 = 0$ for simplicity.

We learn the parameters of the transformer by minimizing the following empirical loss over $B$ independent prompts:

$$\widehat{L}(\theta) = \frac{1}{2B} \sum_{\tau=1}^{B} \left\| f_{\text{LSA}}(E_\tau; \theta)_{[:,-1]} - (0_d, 0, w_\tau, 1) \right\|_{\ell_2}^2 \tag{3.3}$$

We consider the behavior of gradient descent-trained networks over the population loss induced by the limit of infinite training prompts:

$$L(\theta) = \lim_{B \to \infty} \widehat{L}(\theta) = \frac{1}{2} \mathbb{E}_{w_\tau, x_{\tau,1}, \dots, x_{\tau,n}} \left( \left\| f_{\text{LSA}}(E_0; \theta)_{[:,-1]} - (0_d, 0, w_\tau, 1) \right\|_{\ell_2}^2 \right) \tag{3.4}$$

Our first result shows that with suitable initialization and step size, gradient descent converges to a global minimum of $L(\theta)$, which we explicitly characterize.

**Theorem 3.1** *Consider the linear self-attention network over the population loss* (3.4) *with initialization*

$$V(0) = \begin{bmatrix} 0 & 0 & 0 & 0 \\ 0 & 0 & 0 & 0 \\ V_{31}(0) & 0 & 0 & 0 \\ 0 & 0 & 0 & 0 \end{bmatrix}, \qquad W(0) = \begin{bmatrix} 0 & 0 & cI & 0 \\ 0 & 0 & 0 & -c \\ 0 & 0 & 0 & 0 \\ 0 & 0 & 0 & 0 \end{bmatrix}$$

*for some real-valued c. Also define*

$$\Gamma := \left(1 + \frac{1}{n}\right)\Lambda + \frac{1}{n}\text{tr}(\Lambda)I_d \in \mathbb{R}^{d \times d}. \tag{3.5}$$

*We run gradient descent on the population loss with constant step size $\eta \leq 1/(c^2 \|\Gamma\|_{\text{op}})$. We also fix $W_{24}(t) = -c$. The gradient descent converges to a global minimum of the loss given by*

$$V_* = \begin{bmatrix} 0 & 0 & 0 & 0 \\ 0 & 0 & 0 & 0 \\ -\frac{\Gamma^{-1}}{c} & 0 & 0 & 0 \\ 0 & 0 & 0 & 0 \end{bmatrix}, \qquad W_* = \begin{bmatrix} 0 & 0 & cI & 0 \\ 0 & 0 & 0 & -c \\ 0 & 0 & 0 & 0 \\ 0 & 0 & 0 & 0 \end{bmatrix}. \tag{3.6}$$

Note that chain-of-thought reasoning is not employed during training; however, as we discuss in the following section, the model engages in chain-of-thought reasoning at test time. In contrast, (Huang et al., 2025a, Theorem 3.1) consider the setting of isotropic Gaussian features ($\Lambda = I$) and incorporate chain-of-thought reasoning during training by generating intermediate steps through gradient updates on the linear regression objective. Also, the result of (Zhang et al., 2024, Theorem 4.1) does not apply to our setting, since it works with a different embedding and trains the model by minimizing the expected prediction loss function.

## 3.2 TEST TIME CHAIN-OF-THOUGHT

During test time, we observe a test prompt of the form $P = (x_1, \langle w_{\text{test}}, x_1 \rangle, \dots, x_m, \langle w_{\text{test}}, x_m \rangle)$ of possibly different length than the training prompts, and $w_{\text{test}}$ may be never seen before. We let the transformer to generate $k$ steps before it outputs the final prediction $w_k$ of the ground truth $w_{\text{test}}$. Specifically, we let $E_i$ be the embedding at the $i$-th step of generation, and have $f_{\text{LSA}}(E_i)[d+2 : 2d+1, -1]$ as the prediction of the next link in the chain. We then append it to the current embedding, as follows:

$$E_i = \begin{bmatrix} x_1 & \cdots & x_m & 0 & 0 & \dots & 0 \\ y_1 & \cdots & y_m & 0 & 0 & \dots & 0 \\ 0 & \cdots & 0 & w_0 & w_1 & \dots & w_i \\ 0 & \cdots & 0 & 1 & 1 & \dots & 1 \end{bmatrix} := \begin{bmatrix} X_{\text{test}} & 0 & 0 & \dots & 0 \\ y_{\text{test}} & 0 & 0 & \dots & 0 \\ 0_{d \times n} & w_0 & w_1 & \dots & w_i \\ 0_{1 \times n} & 1 & 1 & \dots & 1, \end{bmatrix} \tag{3.7}$$

with $w_i := f_{\text{LSA}}(E_{i-1})_{[d+2:2d+1,-1]}$. The final prediction is given by $w_{k+1}$. In our next proposition, we give an explicit characterization of the recursive updates of $w_i$.

**Proposition 3.2** *Consider the LSA model with parameters $V_*$ and $W_*$ given by* (3.6) *and assume a test prompt of the form $P = (x_1, \langle w_{\text{test}}, x_1 \rangle, \dots, x_m, \langle w_{\text{test}}, x_m \rangle)$. Initializing the test time CoT with $w_0 = 0$, we have*

$$w_{i+1} = w_i - \frac{1}{m}\Gamma^{-1} X_{\text{test}} X_{\text{test}}^\top (w_i - w_{\text{test}}), \tag{3.8}$$

*where $X_{\text{test}} = [x_1 | \dots | x_m] \in \mathbb{R}^{d \times m}$. Therefore, the final output (after $k$ step of generation) is given by*

$$w_{k+1} = \left(I - \left(I - \frac{1}{m}\Gamma^{-1} X_{\text{test}} X_{\text{test}}^\top\right)^k\right) w_{\text{test}}. \tag{3.9}$$

**Remark 3.3** *Consider the quadratic loss* $\ell(w) := \frac{1}{2m} \left\| y_{\text{test}} - X_{\text{test}}^{\mathsf{T}} w \right\|_{\ell_2}^2$, *with* $y_{\text{test}} = X_{\text{test}}^{\mathsf{T}} w_{\text{test}}$. *The gradient of the loss is given by* $\nabla \ell(w) = -\frac{1}{m} X_{\text{test}} (y_{\text{test}} - X_{\text{test}}^{\mathsf{T}} w) = \frac{1}{m} X_{\text{test}} X_{\text{test}}^{\mathsf{T}} (w - w_{\text{test}})$, *and the expected Hessian is given by* $\mathbb{E}[\nabla^2 \ell(w)] = \mathbb{E}[\frac{1}{m} X_{\text{test}} X_{\text{test}}^{\mathsf{T}}] = \Lambda$. *Treating* $\Gamma$, *given by* (3.5), *as a regularized form of* $\Lambda$, *the update* (3.8) *is (pseudo-) Newton's method for optimizing the loss.*

### 3.4 HARDNESS OF A TASK

We define a task by the covariance matrix of its features ($\Lambda$), so different tasks have different features covariances and for each task, we have many prompts with features generated from $\mathsf{N}(0, \Lambda)$, but each with its own $w_\tau$. Now suppose we perform direct in-context learning on a task and then use it to predict labels on queries from the same task (without CoT). Our next result will bound the expected estimation error and we use that to define a measure of task hardness.

**Theorem 3.3** *Consider the LSA model with parameters* $V_*$ *and* $W_*$ *and assume a test prompt is of the form* $P = (x_1, \langle w_{\text{test}}, x_1 \rangle, \dots, x_m, \langle w_{\text{test}}, x_m \rangle)$. *Initializing the in-context learning with* $w_0 = 0$, *the estimate of* $w$ *will be given by* $\hat{w} = \frac{1}{n} \Gamma^{-1} X_{\text{test}} X_{\text{test}}^{\mathsf{T}} w$ *with* $X_{\text{test}} = [x_1 | \dots | x_m] \in \mathbb{R}^{d \times m}$. *We have*

$$\mathbb{E}_{X_{\text{test}}}(\|\hat{w} - w_{\text{test}}\|^2) \leq w_{\text{test}}^{\mathsf{T}} \left( \frac{1}{n^2} (I + \text{tr}(\Lambda)\Lambda^{-1})^2 + \frac{1}{m}(I + \text{tr}(\Lambda^{-1})\Lambda) \right) w_{\text{test}}$$

*where the expectation is with respect to* $X_{\text{test}}$. *Taking expectation with respect to* $w_{\text{test}} \sim \mathsf{N}(0, I)$, *we obtain*

$$\mathbb{E}(\|\hat{w} - w_{\text{test}}\|^2) \leq \frac{d}{n^2} \left( 1 + \frac{\text{tr}(\Lambda)}{\lambda_{\min}(\Lambda)} \right)^2 + \frac{d}{m} \left( 1 + \frac{\text{tr}(\Lambda)}{\lambda_{\min}(\Lambda)} \right). \tag{3.10}$$

Based on the above result, we define the hardness of a task, with features covariance $\Lambda$, via the following measure:

$$\mathsf{Hard}(\Lambda) := \frac{\text{tr}(\Lambda)}{\lambda_{\min}(\Lambda)}. \tag{3.11}$$

Note that it is invariant to scaling of $\Lambda$ and would be higher if $\Lambda$ has some small eigenvalue as more data is needed to learn these directions. Our next results bound the expected estimation error under CoT during test time.

**Theorem 3.4** *Consider the setting of Theorem 3.3 and let* $w_{k+1}$ *be the model estimate for the target task after generating* $k$ *steps during test time. Also suppose that* $m = \Omega(k^2 d)$ *and that eigenvalues of* $\Lambda$ *are upper and lower bounded by positive constants. We have*

$$\mathbb{E}(\|w_{k+1} - w_{\text{test}}\|_{\ell_2}^2) \leq \text{tr}((I - \Gamma^{-1}\Lambda)^{2k})(1 + O(k\sqrt{d/m}))$$

*where the expectation is with respect to* $X_{\text{test}} = [x_1 | \dots | x_m]$ *and* $w_{\text{test}} \sim \mathsf{N}(0, I)$.

**Corollary 3.5** *Under the setting of Theorem 3.4, and letting* $\lambda_{\min}(\Lambda) > 0$ *be the minimum eigenvalue of* $\Lambda$ *we have*

$$\mathbb{E}(\|w_{k+1} - w_{\text{test}}\|_{\ell_2}^2) \leq d \left( 1 + \frac{n}{1 + \mathsf{Hard}(\Lambda)} \right)^{-2k} (1 + o(1)).$$

The above corollary is also consistent with our measure of hardness: the estimation error of $w_{k+1}$ increases with $\mathsf{Hard}(\Lambda)$. In addition, if we want to get the estimation error below some target level $\varepsilon$, harder tasks require longer CoT at test time (larger $k$).

Note that in Corollary 3.5, it was assumed that $\Lambda$ is full rank. If $\Lambda$ is rank deficient (that is the features are coming from a subspace of lower dimension), then one cannot estimate $w_{\text{test}}$ along those directions, as we do not see any information about them during the process. This of course is not an issue if the prompts during test time are coming from the same task, as those directions do not contribute to the predictions. In these cases, by restricting to the relevant subspace, hardness of the task can be defined similarly where $\lambda_{\min}(\Lambda)$ is the minimum "non-zero" eigenvalue of $\Lambda$.

An interpretation of the hardness measure is that each eigenvector of $\Lambda$ corresponds to a specific skill needed for solving examples from that task, with the corresponding eigenvalues indicating the strength of those skills. An easy task is one that relies on a few dominant skills (a small number of nonzero eigenvalues of similar magnitude), while a hard task draws on many skills, reflected in a long-tailed spectrum. The proposed measure captures this intuition quantitatively.

**Remark 3.5 Test-time scaling.** Our result in Corollary 3.5 provides test time scaling for our ICL setting. Note that the computational complexity during test time increases as we allow for more steps of thinking; Specifically, it is $O(kd^2)$ as the matrix $I - \frac{1}{m}\Gamma^{-1}X_{\text{test}}X_{\text{test}}^\top$ can be computed once, and each step of thinking involves multiplying it with the current estimate. Our result also captures the role of $\lambda_{\min}$, $\text{tr}(\Lambda)$ and the prompts length $n$ during training and the features dimension $d$ in shaping the test time scaling law. Another observation is that at any fixed test error, by increasing $k$ we can decrease the length of prompts during training. In Figure 1, we illustrate test-time scaling for several choices of prompt lengths ($n$) and task hardness.

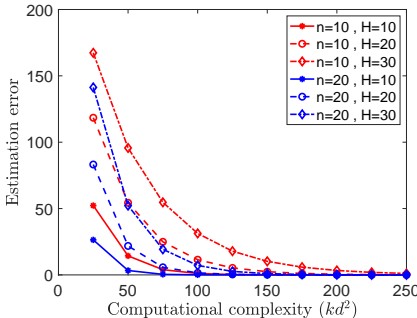

Figure 1: Test-time scaling for the in-context learning. Here, $n$ is the number of in-context examples (context length) in training prompts, and $H$ is the task hardness.

## 4 TASK SELECTION FOR TRAINING

We consider a set of $T$ tasks with corresponding covariances $\Lambda_1, \ldots, \Lambda_T$. Similar to previous sections we draw infinite prompts ($B \to \infty$) but here each prompt is selected from task $i$ with probability $\pi_i \geq 0$, where $\sum_i \pi_i = 1$. The goal of this section is to derive optimal choice of $\{\pi_i\}_{i=1}^T$ and build insights about this choice.

**Theorem 4.1** *Consider the linear self-attention network and the population loss* (3.4) *under the multi-task setting, with the same initialization given in Theorem 3.1. Redefine $\Gamma$ as follows:*

$$\Gamma := \frac{n-1}{n}\sum_{\ell \in [T]} \Lambda_\ell \pi_\ell + \frac{1}{n}\Big(2\sum_{\ell \in [T]} \Lambda_\ell^2 \pi_\ell + \sum_{\ell \in [T]} \text{tr}(\Lambda_\ell)\Lambda_\ell \pi_\ell\Big)\Big(\sum_{\ell \in [T]} \Lambda_\ell \pi_\ell\Big)^{-1}. \quad (4.1)$$

*Then a similar statement to Theorem 3.1 holds true.*

Consider a target task with covariance $\Sigma$ (which can be different from any of the tasks during training). For a prompt from it, $P = (x_1, \langle w_{\text{test}}, x_1 \rangle, \ldots, x_m, \langle w_{\text{test}}, x_m \rangle)$, we let $X_{\text{test}} = [x_1| \ldots |x_m] \in \mathbb{R}^{d \times m}$ and $\widehat{\Sigma} := \frac{1}{m}X_{\text{test}}X_{\text{test}}^\top$. Initializing with $w_0 = 0$ and allowing a chain-of-thought of length $k$, (A.9) implies that the LSA estimate of $w_{\text{test}}$ is $w_{k+1} = (I - (I - \Gamma^{-1}\widehat{\Sigma})^k)w_{\text{test}}$. Therefore,

$$\mathbb{E}(\|w_{k+1} - w_{\text{test}}\|^2) = \mathbb{E}(\|(I - \Gamma^{-1}\widehat{\Sigma})^k w_{\text{test}}\|^2) = \mathbb{E}[\text{tr}((I - \widehat{\Sigma}\Gamma^{-1})^k)(I - \Gamma^{-1}\widehat{\Sigma})^k)], \quad (4.2)$$

where the second step is by taking expectation with respect to $w_{\text{test}}$. In the next proposition, we derive a *prompt instance independent* upper bound for the estimation error in terms of the population covariance $\Sigma$.

**Proposition 4.2** *Suppose that $m = \Omega(k^2 d)$. Then,*

$$\mathbb{E}[\text{tr}((I - \widehat{\Sigma}\Gamma^{-1})^k)(I - \Gamma^{-1}\widehat{\Sigma})^k)] \leq \text{tr}(\Gamma)\text{tr}(\Gamma^{-1})\text{tr}((I - \Gamma^{-1/2}\Sigma\Gamma^{-1/2})^{2k})(1 + o(1)). \quad (4.3)$$

The optimal choice of tasks selection probabilities is the one that minimizes the expected estimation error during test time given by (4.2). We instead use the upper bound given by Proposition 4.2 and focus on the term $\text{tr}((I - \Gamma^{-1/2}\Sigma\Gamma^{-1/2})^{2k})$ in (4.3), which captures the effect of thinking and is the dominant term with an exponential rate. This results in the following optimization for choosing task selection probabilities:

$$\min_{\pi_\ell, \ell \in [m]} \quad \mathbb{E}[\text{tr}((I - \Gamma^{-1/2}\Sigma\Gamma^{-1/2})^{2k})] \tag{4.4}$$

$$\text{subject to} \sum_{\ell \in [T]} \pi_\ell = 1, \quad \pi_\ell \geq 0, \ \forall \ell \in [T]$$

**Remark 4.1 When is the test time thinking useful?** We observe that the effect of thinking at inference time is captured by the term $\text{tr}((I - \Gamma^{-1/2}\Sigma\Gamma^{-1/2})^{2k})$. Depending on the eigenvalues of $\Gamma^{-1/2}\Sigma\Gamma^{-1/2}$, this term may shrink or grow as $k$ increases. Intuitively, if each eigenvector of $\Sigma$ (representing the skills required at test time) is sufficiently represented in the training data—so that $\Gamma$ is strong along that direction and $\Gamma^{-1/2}\Sigma\Gamma^{-1/2}$ remains small—then additional thinking improves performance. In contrast, if some task-relevant directions are underrepresented in the training data, and thus not well learned by the model, increasing the amount of test-time thinking can degrade performance, effectively leading to overthinking.

## 4.2 Optimal choice of task selection probabilities

We next analyze the optimization (4.4) to argue that choosing a diverse, relevant and hard set of tasks during training results in best performance for test-time scaling.

**Diversity.** A key observation is that we must select a *diverse* set of tasks so that the spectrum of $\Gamma$ adequately covers all directions in the target covariance $\Sigma$. Failing to do so causes $\Gamma^{-1/2}\Sigma\Gamma^{-1/2}$ to be large along uncovered directions, resulting in higher test error that may further amplify with additional reasoning steps.

**Relevance.** Another important notion is the *relevance* of the selected tasks to the target task. Recall the expression of $\Gamma$ given by (4.1). When $d \ll n$, and noting that the eigenvalues of $\Lambda$ are $O(1)$, $\Gamma$ can be replaced by $\tilde{\Gamma} := \sum_{\ell \in [T]} \Lambda_\ell \pi_\ell$, which is a convex combination of $\{\Lambda_\ell\}_{\ell \in [m]}$. Hence, minimizing $\text{tr}((I - \Gamma^{-1/2}\Sigma\Gamma^{-1/2})^{2k})$ in effect corresponds to approximating $\Sigma$ with a convex combination of the task covariance matrices and so tasks which place high weight on directions well represented in $\Sigma$ (i.e. relevant ones) are desirable.

**Hardness.** The other factor in task selection is the hardness of tasks. We argue that when the target task is hard (as is often the case where models are compared on difficult benchmarks), our proposal favors selecting hard tasks during training. Without loss of generality, by scaling features we can assume that $\text{tr}(\Lambda_\ell) = 1, \forall \ell$ and $\text{tr}(\Sigma) = 1$. With this normalization the hardness of task is captured by the minimum eigenvalue of the corresponding covariance matrix. Now, invoking the test error given by $\text{tr}((I - \Gamma^{-1/2}\Sigma\Gamma^{-1/2}))$, the absolute error along minimum eigenvectors of $\Sigma$ contribute more towards the error. Given that the target task is a hard one, $\sigma_{\min}(\Sigma)$ is small and in the next proposition we show that to estimate $\Sigma$ well on this direction by a convex combination of available tasks, we need to select some hard tasks (those with small minimum eigenvalue).

**Proposition 4.3** *Suppose that $|\sigma_{\min}(\Gamma) - \sigma_{\min}(\Sigma)| \leq \varepsilon$ and define $D := \{\ell \in [T], \ \sigma_{\min}(\Lambda_\ell) \leq 4(\varepsilon + \sigma_{\min}(\Sigma))\}$. Note that $D$ corresponds to tasks with small minimum eigenvalues (hard tasks), since both $\varepsilon$ and $\sigma_{\min}(\Sigma)$ are small. Then, $\sum_{\ell \in D} \pi_\ell \geq 1/2$. In words, at least $1/2$ of task selection probabilities are on hard tasks.*

**Further simplification of task selection procedure.** Note that the optimization problem (4.4) is inherently nonconvex, which motivates us to turn to simplifications that transform it into a convex and tractable form for large-scale problems. We make two modifications: 1) As discussed before when $d \ll n$ and since the the eigenvalues of $\Lambda$ are $O(1)$, $\Gamma$ can be well approximated by $\tilde{\Gamma} := \sum_{\ell \in [T]} \Lambda_\ell \pi_\ell$, which is a convex combination of $\{\Lambda_\ell\}_{\ell \in [m]}$. 2) The objective in (4.4) seeks to make $\Gamma^{-1/2}\Sigma\Gamma^{-1/2}$ close to the identity matrix. Instead, we minimize $\|I - \Sigma^{-1}\Gamma\|_F^2 \approx \|I - \Sigma^{-1}\tilde{\Gamma}\|_F^2$, which pursues the same goal but through a different formulation. With these consideration, we propose the following

alternative optimization for choosing task selection probabilities $\{\pi_\ell\}_{\ell \in [T]}$:

$$\min_{\{\pi_\ell\}_{\ell \in [T]}} \quad \left\| I - \Sigma^{-1} \sum_{\ell \in [T]} \Lambda_\ell \pi_\ell \right\|_F^2 \tag{4.5}$$

$$\text{subject to} \sum_{\ell \in [T]} \pi_\ell = 1, \quad \pi_\ell \geq 0, \ \forall \ell \in [T]$$

This is a quadratic optimization problem and can be efficiently solved at scale.

## 5 EXPERIMENTS

In this section, we conduct experiments to validate our theoretical results.

**Setting.** We conduct experiments in two settings. First, we consider a transformer with a single linear self-attention (LSA) to confirm our theory. Then, we consider large, nonlinear transformer architecture namely GPT-2 to validate our conclusions. In both cases, the data distribution follows our in-context weight prediction task in Sec. 3.1, where $x_{\tau,i} \sim \mathsf{N}(0, \Lambda)$, $w_\tau \sim \mathsf{N}(0, I_d)$. We choose the token dimensions $d = 10$. During inference, we let the model to output multiple steps before returning the final predicted weight vector. At each step $i$ we concatenate the embedding with $[0_d, \hat{w}_i, 1]$ as in Eq. (3.7) and input the concatenated embedding matrix to the model. The predicted $w_k$ will be returned after $k$ steps of CoT. We report the average results and error bars over 10 runs.

**Transformers with a single linear self-attention (LSA).** We train the transformer architecture in Eq. (3.2) on the synthetic data generated as described above. We generate 5000 examples, use a batch size $B = 1000$ and run Adam with learning rate $\eta = 0.001$ for $= 1000$ epochs. For the results reported in the main paper we follow our theoretical setting in Sec. 3.1. That is, we initialize transformer weights $(V(0), W(0))$ according to Theorem 3.1 where $V31(0)$ is set randomly with entries independently and uniformly drawn from $[0, 1]$, and we set $c = 1$. We do not perform CoT during training.

We report additional results with random initialization and training with CoT in Appendix B.

**Large, nonlinear transformer architectures.** We use a decoder-only Transformer architecture (Vaswani et al., 2017) from the GPT-2 family (Radford et al., 2019), consisting of 12 layers, 8 attention heads, and a 256-dimensional embedding space. In total the model contains 9.5M parameters.

This architecture takes as input a sequence of vectors in its embedding space and predicts the weight vector within the same space. We apply this architecture to prompts of form $(x_{\tau,1}, y_{\tau,1}, \cdots, x_{\tau,m}, y_{\tau,m}, w_0, 1)$ in the following manner. In line with (Garg et al., 2022), we map each $y_{\tau,i}$ to the same dimension as $x_{\tau,i}$ by appending zeros, and map $x_{\tau,i}, y_{\tau,i}$ into the latent embedding space of the Transformer through a (learnable) linear transformation. We get the predicted $w_\tau$ as the model output. Similarly, we map the model output, i.e., $w_\tau$ from the latent embedding space of the Transformer to a $d$-dimensional vector through another (learnable) linear transformation. Training is performed with a batch size of 64 over $25k$ total steps. The model weights are randomly initialized, and CoT is applied during both training and inference. We use curriculum learning (Garg et al., 2022) to speed up training.

**Larger test-time compute reduces the requirement on training-time compute.** Fig 2a, 2c show the test error vs length of CoT ($k$). For the LSA model, we use $n = 10, 20, 30$ and for GPT-2 we use $n = 20, 30, 40$. We see that by increasing the test-time compute ($k$), we can decrease the length of prompts $n$ during training to get a similar test error.

**When more thinking hurts.** For training, we sample prompt inputs from $\mathsf{N}(0, \Lambda)$ where $\Lambda$ is a skewed covariance matrix with eigenbasis chosen uniformly at random and $i$-th eigenvalue proportional to $1/i$. For test, we sample prompt inputs from $\mathsf{N}(0, I_d)$. We normalize the inputs so that their expected squared norm is equal to that of inputs encountered during training. Fig 2b, 2d show that when some of the directions of the target task are not sufficiently present in the training data, allowing for more thinking during test time would hurt the performance. An interesting observation is that when the model is in overthinking regime (Fig 2b, 2d) larger prompt length $n$ yields a higher test loss, while when the model is not overthinking, larger $n$ reduces the test loss (Fig 2a, 2c).

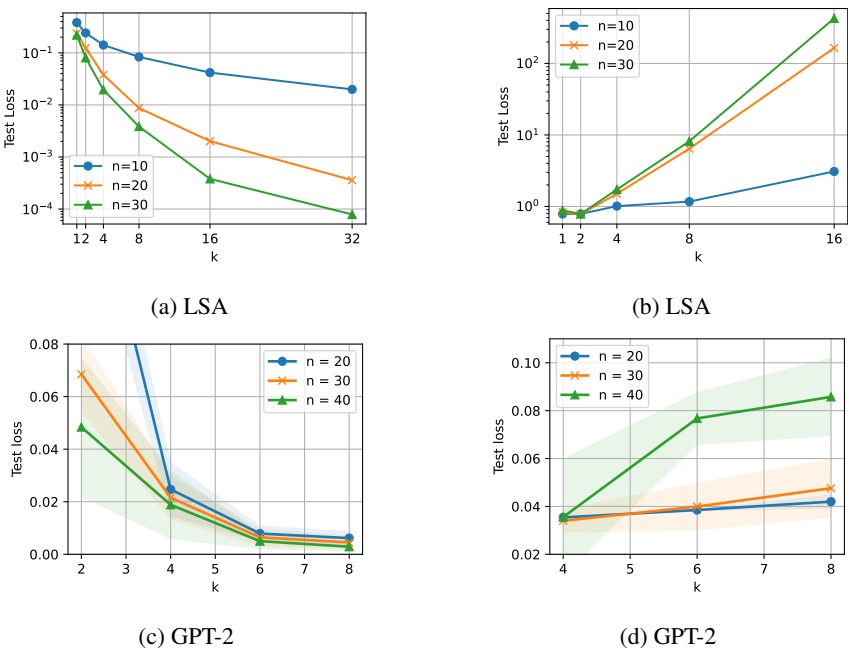

(a) LSA

(b) LSA

(c) GPT-2

(d) GPT-2

Figure 2: More test-time compute reduces training-time requirements for (a) one-layer transformer and (c) GPT-2. However, insufficient task coverage in training data makes longer CoTs harmful for (b) one-layer transformer and (d) GPT-2. For GPT-2, the errorbars are std of 10 runs. For LSA, std is negligible as we start from the fixed initialization in Eq. (3.6). Same value for $n$ is used in training and test.

## 5.1 TASK SELECTION

We design an experiment to illustrate that our method prioritizes diverse and hard tasks. We consider a multi-task setup with four task types, where each type is defined by two parameters $\alpha$ and $B$. These parameters respectively control the decay rate and the support size of the eigenvalues. Specifically, eigenvalues are proportional to $i^{-\alpha}$ for $i \in [B]$ and zero elsewhere. The positions of nonzero eigenvalues are uniformly shuffled within $[d]$, and the eigenvalues are scaled to have unit sum. Here, $B$ captures task diversity, while $\alpha$ captures the task hardness (with larger $\alpha$ producing smaller nonzero eigenvalues, corresponding to harder tasks according to measure (3.11)).

The four training task types are: Easy-Short ($\alpha = 0.2, B = 20$), Hard-Short ($\alpha = 0.8, B = 20$), Easy-Long ($\alpha = 0.2, B = 100$), and Hard-Long ($\alpha = 0.8, B = 100$). The target task is set with $\alpha = 0.8$ and $B = d = 1000$. We generate 50 tasks of each type by randomizing the eigenbases and the support of eigenvalues. We then solve the quadratic optimization problem (4.5) to obtain task selection probabilities $\pi_\ell$, for $\ell = 1, \ldots, 200$. Fig 3a displays these probabilities, colored by task type, with solid lines indicating their average per type. As shown, harder and more diverse tasks receive higher selection probabilities, while easier, more concentrated tasks are weighted lower. Fig 3b further plots selection probability versus task hardness, confirming that harder tasks are indeed favored, consistent with our theoretical analysis in Section 5.1.

To demonstrate the improvement achieved by our task selection procedure, Appendix B.1 reports simulation results showing test error as a function of the test-time thinking length ($k$) for different task selection strategies. The findings indicate that our optimal task selection procedure effectively prevents overthinking.

## 5.2 EVALUATION ON REAL REASONING BENCHMARKS

Finally, to demonstrate the validity of our conclusions on real reasoning tasks, we train Qwen 2.5-7B-Instruct on the OMEGA dataset (Sun et al., 2025). We chose two tasks from the OMEGA dataset, namely GCD and polynomial root reasoning. These tasks are designed such that training on one does not benefit the performance on the other. We fine-tune the base model (Qwen-Base) with RL

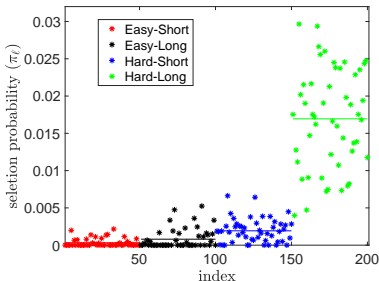

(a) Selection probabilities for different task types

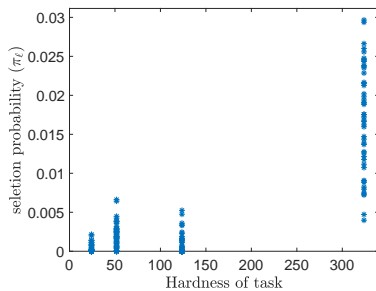

(b) Selection probabilities vs. task hardness

Figure 3: Task selection in a multi-task setup (a) Each color corresponds to a task type with solid lines indicating the average selection probability per type. As we observe harder and more diverse tasks receive higher selection probabilities, while easier, more concentrated tasks are weighted lower (b) Task selection probabilities versus task hardness. As we see harder task are favored in the selection.

separately on the training data of GCD and Poly. We call these models Qwen-GCD and Qwen-Poly. We evaluate both models on the test data of GCD. Table 1 shows that, as expected, for the harder tasks that require longer reasoning, all models have a lower performance. However, we see that while shorter test-time thinking (CoT length less than 1k characters) yields a much better performance (+44.69%) on GCD for Qwen-GCD compared to Qwen-Base, it yields a slightly lower performance on GCD for Qwen-Poly, compared to Qwen-Base (-1.39%). Interestingly, when models reason for longer at test-time (between 1k and 2k characters), Qwen-Poly has a much lower performance (-6.37%) compared to Qwen-Base, while Qwen-GCD outperforms Qwen-Base by 11.2%. This confirms our theoretical results that when training and test data are aligned, more thinking helps. But, insufficient task coverage in training data makes longer test-time compute harmful.

Table 1: Average accuracy on GCD for Qwen2.5-7B Instruct (Base), Base model fine-tuned on CGD (Qwen-GCD) and Base model fine-tuned on Poly (Qwen-Poly). For all the models, the accuracy on examples that require longer CoT is lower (compare the second column to the first column). This confirms that examples that require longer CoT are generally more difficult. The % in () shows the fraction of test data with the corresponding test-time CoT length. Notably, shorter CoTs (0-1k) considerably improves the performance of Qwen-GCD (75% versus 30.39%) and slightly harms the performance of Qwen-Poly (29% versus 30.39%). Longer CoTs improve the performance of Qwen-GCD (38.4% versus 27.2%) and significantly harm the performance of Qwen-Poly (20.83% versus 27.2%).

| CoT length | [0, 1k) | [1k, 2k] |
|---|---|---|
| Qwen-Base | 30.39% (30% data) | 27.2% (70% data) |
| Qwen-GCD | 75% (15% data) | 38.4% (85% data) |
| Qwen-Poly | 29% (32% data) | 20.83% (68% data) |

## 6 CONCLUSION

In this work, we provided a theoretical and empirical framework for understanding in-context learning in transformers, showing that chain-of-thought prompting at test time enables models to emulate multi step (pseudo)-Newton's method. By introducing a principled notion of task hardness based on features covariance spectrum, we derived scaling laws that clarify how test-time compute, context length, and task diversity interact. We proposed an optimal strategy for task selection in a multi-task training that shows training on a diverse, relevant and hard set of tasks during training results in best performance for test-time scaling. We also validated our findings on both linear self-attention models and GPT-2. We will conclude by discussing some limitations of our work which pave the way for future directions of work. Our theoretical analysis in this work is limited to linear regression tasks and single-layer linear self-attention, and an important direction for future work is extending these results to nonlinear data generation settings and transformers with nonlinear activations.

ACKNOWLEDGMENTS

AJ was supported in part by the Sloan fellowship in mathematics, the NSF Award DMS-2311024, an Amazon Faculty Research Award, an Adobe Faculty Research Award and an iORB grant form USC Marshall School of Business. BM was supported in part by the NSF CAREER Award 2146492, NSF-Simons AI Institute for Cosmic Origins (CosmicAI) and NSF AI Institute for Foundations of Machine Learning (IFML).

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

# A    PROOFS OF THEOREMS AND TECHNICAL LEMMAS

## A.1    PROOF OF THEOREM 3.1

**Lemma A.1** *Assume an initialization of the form*

$$
V(0) = \begin{bmatrix} 0 & 0 & 0 & 0 \\ 0 & 0 & 0 & 0 \\ V_{31}(0) & 0 & 0 & 0 \\ 0 & 0 & 0 & 0 \end{bmatrix}, \qquad
W(0) = \begin{bmatrix} 0 & 0 & cI & 0 \\ 0 & 0 & 0 & -c \\ 0 & 0 & 0 & 0 \\ 0 & 0 & 0 & 0 \end{bmatrix}.
$$

*When the linear transformer is trained under gradient descent. Then $V(t)$ and $W(t)$ have the following form:*

$$
V(t) = \begin{bmatrix} 0 & 0 & 0 & 0 \\ 0 & 0 & 0 & 0 \\ V_{31}(t) & 0 & 0 & 0 \\ 0 & 0 & 0 & 0 \end{bmatrix}, \qquad
W(t) = \begin{bmatrix} 0 & 0 & cI & 0 \\ 0 & 0 & 0 & W_{24}(t) \\ 0 & 0 & 0 & 0 \\ 0 & 0 & 0 & 0 \end{bmatrix}.
$$

*with $V_{31}(t) \in \mathbb{R}^{d \times d}$ and $W_{24}(t) \in \mathbb{R}$.*

Lemma A.1 is very similar to (Huang et al., 2025a, Lemma C.2). The difference is that here the features covariance is non-identity, while we do not do CoT during training.

Given that several blocks of $V(t)$ and $W(t)$ remain zero across the gradient updates, we can reduce the loss function in a simpler form. Define the shorthand $\tilde{V}(t) := V_{31}(t) \in \mathbb{R}^{d \times d}$ and $w(t) := W_{24}(t) \in \mathbb{R}$. Invoking (A.3) we can rewrite the loss as follows:

$$
\begin{aligned}
L(\theta(t)) &= \frac{1}{2}\mathbb{E}\left( \left\| f_{\text{LSA}}(E_\tau; \theta(t))_{[:,-1]} - (0_d, 0, w_\tau, 1) \right\|_{\ell_2}^2 \right) \\
&= \frac{1}{2}\mathbb{E}\left( \left\| \begin{bmatrix} 0 \\ 0 \\ \hat{w}_0 \\ 1 \end{bmatrix} + \frac{1}{n} \begin{bmatrix} 0_d \\ 0 \\ V_{31}(t) X X^\top (c\hat{w}_0 + W_{24}(t) w_\tau) \\ 0 \end{bmatrix} - \begin{bmatrix} 0 \\ 0 \\ w_\tau \\ 1 \end{bmatrix} \right\|_{\ell_2}^2 \right) \\
&= \frac{1}{2}\mathbb{E}\left( \left\| (\tilde{V}(t) w(t) \hat{\Lambda} - I) w_\tau \right\|_{\ell_2}^2 \right),
\end{aligned} \tag{A.1}
$$

with $\hat{\Lambda} := X X^\top / n$. As we see $w(t)$ does not provide additional degree of freedom in minimizing the loss since it appears as the term $\tilde{V}(t) w(t)$. This clarifies fixing $w(t) = -c$ along the gradient updates.

We then have

$$
L(\theta) = \frac{1}{2}\mathbb{E}\left[ \text{tr}(c^2 \tilde{V} \hat{\Lambda}^2 \tilde{V}^\top + I + 2c \tilde{V} \hat{\Lambda}) \right]
$$

**Lemma A.2** *Let $X \in \mathbb{R}^{d \times n}$ with columns drawn i.i.d from $\mathsf{N}(0, \Lambda)$. For any deterministic matrix $A$ we have*

$$
\mathbb{E}\left( \frac{X X^\top}{n} A \frac{X X^\top}{n} \right) = \frac{n-1}{n} \Lambda A \Lambda + \frac{1}{n}\left( \Lambda(A + A^\top)\Lambda + \text{tr}(\Lambda A)\Lambda \right).
$$

Using Lemma A.2 we have

$$
\mathbb{E}(\hat{\Lambda}^2) = \frac{n+1}{n}\Lambda^2 + \frac{1}{n}\text{tr}(\Lambda)\Lambda := \Gamma\Lambda.
$$

Therefore,

$$
L(\theta) = \frac{c^2}{2}\text{tr}\left( \tilde{V}\Gamma\Lambda\tilde{V}^\top \right) + \frac{d}{2} + c\,\text{tr}(\tilde{V}\Lambda) \tag{A.2}
$$

This is convex in $\tilde{V}$ and so gradient descent with a fixed step size $\eta \leq 1/L$ converges to its minimizer, if $\nabla_{\tilde{V}}^2 L \preceq LI$. We have

$$
\nabla_{\tilde{V}}^2 L = c^2 \Gamma\Lambda,
$$

so we can take $L = c^2 \|\Gamma\Lambda\|_{\text{op}}$. To find the minimizer of $L(\theta)$, we set its gradient to zero,

$$\frac{c^2}{2}(\Gamma\Lambda\tilde{V}^\top + \tilde{V}\Gamma\Lambda) + c\Lambda = 0\,,$$

which is a continuous-time Lyapunov equation (Note that $\Gamma$ and $\Lambda$ commute and both are symmetric). Hence, it has a unique solution given by $\tilde{V} = -\frac{\Gamma^{-1}}{c}$.

As the final step, we show that $V_*$ and $W_*$ are also a global optimum for the population loss, even without making the specific structure imposed by gradient descent as described in Lemma A.1.

We continue by computing the output of LSA, and recall that $\hat{w}_0 = 0$.

$$V E_\tau \cdot \frac{E_\tau^\top W E_\tau[:,-1]}{n}$$

$$= \frac{1}{n} V \begin{bmatrix} X & 0 \\ y & 0 \\ 0_{d\times n} & 0 \\ 0_{1\times n} & 1 \end{bmatrix} E_\tau^\top W \begin{bmatrix} 0 \\ 0 \\ 0 \\ 1 \end{bmatrix}$$

$$= \frac{1}{n} V \begin{bmatrix} XX^\top & Xy^\top & 0 & 0 \\ yX^\top & yy^\top & 0 & 0 \\ 0_{d\times n} & 0_{d\times 1} & 0_{d\times 1} & 0_{d\times 1} \\ 0_{1\times n} & 0_{1\times n} & 0 & 1 \end{bmatrix}^\top \begin{bmatrix} W_{14} \\ W_{24} \\ W_{34} \\ W_{44} \end{bmatrix}$$

$$= \frac{1}{n} V \begin{bmatrix} XX^\top W_{14} + Xy^\top W_{24} \\ yX^\top W_{14} + yy^\top W_{24} \\ 0 \\ W_{44}(t) \end{bmatrix}$$

Therefore,

$$f_{\text{LSA}}(E_\tau, \theta)[:,-1] - (0_d, 0, w_\tau, 1)^\top$$

$$= \begin{bmatrix} 0 \\ 0 \\ 0 \\ 1 \end{bmatrix} + \frac{1}{n} V \begin{bmatrix} XX^\top W_{14} + Xy^\top W_{24} \\ yX^\top W_{14} + yy^\top W_{24} \\ 0 \\ W_{44} \end{bmatrix} - \begin{bmatrix} 0 \\ 0 \\ w_\tau \\ 1 \end{bmatrix}$$

$$= \frac{1}{n} V \begin{bmatrix} XX^\top W_{14} + XX^\top w_\tau W_{24} \\ w_\tau^\top XX^\top W_{14} + w_\tau^\top XX^\top w_\tau W_{24} \\ 0 \\ W_{44} \end{bmatrix} - \begin{bmatrix} 0 \\ 0 \\ w_\tau \\ 0 \end{bmatrix}$$

The loss function is the expected squared norm of this object. To minimize it, we can make its first, second and last entries zero by setting the corresponding rows in $V$ to zero. This gives us

$$L(\theta)$$

$$\geq \frac{1}{2}\mathbb{E}\left(\left\|\frac{1}{n}(V_{31} + V_{32}w_\tau^\top)XX^\top(W_{14} + W_{24}w_\tau) - w_\tau + \frac{1}{n}V_{34}W_{44}\right\|_{\ell_2}^2\right)$$

$$= \frac{1}{2}\mathbb{E}\left(\left\|V_{31}\hat{\Lambda}W_{14} + \frac{1}{n}V_{34}W_{44} + (V_{31}\hat{\Lambda}W_{24} - I)w_\tau + V_{32}w_\tau^\top\hat{\Lambda}W_{14} + V_{32}w_\tau^\top\hat{\Lambda}W_{24}w_\tau\right\|_{\ell_2}^2\right)$$

$$\geq \frac{1}{2}\mathbb{E}\left(\left\|(V_{31}\hat{\Lambda}W_{24} - I)w_\tau + V_{32}w_\tau^\top\hat{\Lambda}W_{14}\right\|_{\ell_2}^2\right)$$

$$= \frac{1}{2}\mathbb{E}\left(\left\|(V_{31}W_{24}\hat{\Lambda} + V_{32}W_{14}^\top\hat{\Lambda} - I)w_\tau\right\|_{\ell_2}^2\right)\,,$$

where the penultimate step holds since the cross term is an odd function of $w_\tau$ and so its expectation is zero. The other eliminated term is squared and so non-negative. In the last step we used that $w_\tau^\top\hat{\Lambda}W_{14}$ is scalar and so can be replaced by its transpose. In addition $W_{24}$ is also scalar and can

commute with $\hat{\Lambda}$. Note that $V_{32}$ and $W_{14}$ do not offer any flexibility in minimizing the loss function as their effect can be absorbed in $V_{31}W_{24}$. Therefore, we can set them to zero. Hence,

$$\min_{V,W} L(\theta) \geq \min_{V_{31},W_{24}} \frac{1}{2} \mathbb{E}\left(\left\|(V_{31}W_{24}\hat{\Lambda} - I)w_\tau\right\|_{\ell_2}^2\right).$$

Observe that the right-hand side is of the form (A.1) and hence its global optimum (reached by gradient descent) serves as the global minimum of the loss.

### A.1.1 PROOF OF LEMMA A.1

To prove this lemma, we prove that when the irrelevant blocks are 0, the gradients of the loss remain zero on those blocks and they never update the corresponding parameter block.

We do induction on $t$. Suppose that the claim holds for $t$. We start by computing the output of LSA.

$$VE_\tau \cdot \frac{E_\tau^\top W E_\tau[:,-1]}{n}$$

$$= \frac{1}{n}\begin{bmatrix} 0 & 0 & 0 & 0 \\ 0 & 0 & 0 & 0 \\ V_{31}(t) & 0 & 0 & 0 \\ 0 & 0 & 0 & 0 \end{bmatrix}\begin{bmatrix} X & 0 \\ y & 0 \\ 0_{d\times n} & \hat{w}_0 \\ 0_{1\times n} & 1 \end{bmatrix} E_\tau^\top W \begin{bmatrix} 0 \\ 0 \\ \hat{w}_0 \\ 1 \end{bmatrix}$$

$$= \frac{1}{n}\begin{bmatrix} 0_{d\times n} & 0_d \\ 0_{1\times n} & 0 \\ V_{31}(t)X & 0_d \\ 0_{1\times n} & 0 \end{bmatrix}\begin{bmatrix} X & 0 \\ y & 0 \\ 0_{d\times n} & \hat{w}_0 \\ 0_{1\times n} & 1 \end{bmatrix}^\top \begin{bmatrix} c\hat{w}_0 \\ W_{24}(t) \\ 0 \\ 0 \end{bmatrix}$$

$$= \frac{1}{n}\begin{bmatrix} 0_{d\times n} & 0_d \\ 0_{1\times n} & 0 \\ V_{31}(t)X & 0_d \\ 0_{1\times n} & 0 \end{bmatrix}\begin{bmatrix} cX^\top \hat{w}_0 + W_{24}(t)y^\top \\ 0 \end{bmatrix}$$

$$= \frac{1}{n}\begin{bmatrix} 0_d \\ 0 \\ V_{31}(t)XX^\top(c\hat{w}_0 + W_{24}(t)w_\tau) \\ 0 \end{bmatrix}, \tag{A.3}$$

where we used that $y^\top = X^\top w_\tau^*$. We proceed with calculating the derivatives of the loss:

$$\nabla_V L(\theta(t)) = \frac{1}{2}\nabla_V \mathbb{E}\left(\left\|f_{\text{LSA}}(E_\tau; \theta(t))_{[:,-1]} - (0_d, 0, w_\tau, 1)\right\|_{\ell_2}^2\right)$$

$$= \mathbb{E}\left[\left(f_{\text{LSA}}(E_\tau; \theta(t))_{[:,-1]} - (0_d, 0, w_\tau, 1)\right)E_\tau[:,-1]^\top W^\top \frac{E_\tau E_\tau^\top}{n}\right]$$

$$= \mathbb{E}\left[\left(VE_\tau \cdot \frac{E_\tau^\top W E_\tau[:,-1]}{n} - (0_d, 0, w_\tau - \hat{w}_0, 0)\right)E_\tau[:,-1]^\top W^\top \frac{E_\tau E_\tau^\top}{n}\right] \tag{A.4}$$

We note that

$$VE_\tau \cdot \frac{E_\tau^\top W E_\tau[:,-1]}{n} - (0_d, 0, w_\tau - \hat{w}_0, 0)$$

$$= \frac{1}{n}\begin{bmatrix} 0_d \\ 0 \\ V_{31}(t)XX^\top(c\hat{w}_0 + W_{24}(t)w_\tau) + n(\hat{w}_0 - w_\tau) \\ 0 \end{bmatrix}. \tag{A.5}$$

In addition,

$$E_\tau[:,-1]^\top W^\top \frac{E_\tau E_\tau^\top}{n} = \frac{1}{n}\begin{bmatrix} c\hat{w}_0 & W_{24}(t) & 0 & 0 \end{bmatrix}\begin{bmatrix} XX^\top & Xy^\top & 0 & 0 \\ yX^\top & yy^\top & 0 & 0 \\ 0 & 0 & \hat{w}_0\hat{w}_0^\top & \hat{w}_0 \\ 0 & 0 & \hat{w}_0^\top & 1 \end{bmatrix}$$

$$= \frac{1}{n}\begin{bmatrix} c\hat{w}_0 XX^\top + W_{24}(t)yX^\top & c\hat{w}_0 Xy^\top + W_{24}(t)yy^\top & 0 & 0 \end{bmatrix}. \tag{A.6}$$

Plugging in from (A.5) and (A.6) into (A.4) we get the following structure for the gradient of the loss:

$$\nabla L(\theta(t)) = \begin{bmatrix} 0 & 0 & 0 & 0 \\ 0 & 0 & 0 & 0 \\ \nabla_{V_{31}} L(\theta(t)) & \nabla_{V_{32}} L(\theta(t)) & 0 & 0 \\ 0 & 0 & 0 & 0 \end{bmatrix}, \tag{A.7}$$

where

$$\nabla_{V_{31}} L(\theta(t)) = \frac{1}{n^2} \mathbb{E}\left[\left(V_{31}(t)XX^\top(c\hat{w}_0 + W_{24}(t)w_\tau) + n(\hat{w}_0 - w_\tau)\right)\left(c\hat{w}_0 XX^\top + W_{24}(t)yX^\top\right)\right]$$

$$\nabla_{V_{32}} L(\theta(t)) = \frac{1}{n^2} \mathbb{E}\left[\left(V_{31}(t)XX^\top(c\hat{w}_0 + W_{24}(t)w_\tau) + n(\hat{w}_0 - w_\tau)\right)\left(c\hat{w}_0 Xy^\top + W_{24}(t)yy^\top\right)\right]$$

Recall that we set $\hat{w}_0 = 0$ by which we obtain

$$\nabla_{V_{32}} L(\theta(t)) = \frac{1}{n^2} \mathbb{E}\left((V_{31}(t)W_{24}(t)XX^\top - nI)w_\tau W_{24}(t)yy^\top\right)$$

$$= \frac{W_{24}(t)}{n^2} \mathbb{E}\left((V_{31}(t)W_{24}(t)XX^\top - nI)w_\tau w_\tau{}^\top XX^\top w_\tau\right) = 0$$

since it is an odd function of $w_\tau \sim \mathsf{N}(0, I)$. (Note that the population loss is non-random, due to the expectation in its definition. Since the initial $V(0), W(0)$ are non-random, the trajectory $V(t)$ and $W(t)$ are non-random.)

We next proceed with calculating the gradient with respect to $W$. We have

$$\nabla_W L(\theta(t))$$

$$= \frac{1}{2} \nabla_W \mathbb{E}\left(\left\| f_{\mathrm{LSA}}(E_\tau; \theta(t))_{[:, -1]} - (0_d, 0, w_\tau, 1)\right\|_{\ell_2}^2\right)$$

$$= \frac{1}{n} \mathbb{E}\left[E_\tau E_\tau^\top V^\top \left(f_{\mathrm{LSA}}(E_\tau; \theta(t))_{[:, -1]} - (0_d, 0, w_\tau, 1)\right) E_\tau[:, -1]^\top\right]$$

$$= \frac{1}{n} \mathbb{E}\left[E_\tau E_\tau^\top V^\top \left(VE_\tau \cdot \frac{E_\tau^\top W E_\tau[:, -1]}{n} - (0_d, 0, w_\tau - \hat{w}_0, 0)\right) E_\tau[:, -1]^\top\right]$$

$$= \frac{1}{n^2} \mathbb{E}\left(\begin{bmatrix} XX^\top V_{31}(t)^\top \left(V_{31}(t)XX^\top(c\hat{w}_0 + W_{24}(t)w_\tau) + n(\hat{w}_0 - w_\tau)\right) \\ yX^\top V_{31}(t)^\top \left(V_{31}(t)XX^\top(c\hat{w}_0 + W_{24}(t)w_\tau) + n(\hat{w}_0 - w_\tau)\right) \\ 0 \\ 0 \end{bmatrix} \begin{bmatrix} 0 & 0 & \hat{w}_0 & 1 \end{bmatrix}\right),$$

where the last step follows from the following equation and simple algebraic calculation:

$$E_\tau E_\tau^\top = \begin{bmatrix} XX^\top & Xy^\top & 0 & 0 \\ yX^\top & yy^\top & 0 & 0 \\ 0 & 0 & \hat{w}_0\hat{w}_0^\top & \hat{w}_0 \\ 0 & 0 & \hat{w}_0^\top & 1 \end{bmatrix}, \quad E_\tau[:, -1]^\top = \begin{bmatrix} 0 & 0 & \hat{w}_0 & 1 \end{bmatrix},$$

$$VE_\tau \cdot \frac{E_\tau^\top W E_\tau[:, -1]}{n} - (0_d, 0, w_\tau - \hat{w}_0, 0) = \begin{bmatrix} 0_d \\ 0 \\ V_{31}(t)XX^\top(c\hat{w}_0 + W_{24}(t)w_\tau) + n(\hat{w}_0 - w_\tau) \\ 0 \end{bmatrix}$$

Recalling that $\hat{w}_0 = 0$ we simplify $\nabla_W L$ as follows:

$$\nabla_W L(\theta(t)) = \frac{1}{n^2} \mathbb{E}\left(\begin{bmatrix} XX^\top V_{31}(t)^\top \left(V_{31}(t)W_{24}(t)XX^\top - nI\right)w_\tau \\ yX^\top V_{31}(t)^\top \left(V_{31}(t)W_{24}(t)XX^\top - nI\right)w_\tau \\ 0 \\ 0 \end{bmatrix} \begin{bmatrix} 0 & 0 & 0 & 1 \end{bmatrix}\right). \tag{A.8}$$

We have $\mathbb{E}[XX^\top V_{31}(t)^\top (V_{31}(t)W_{24}(t)XX^\top + nI)w_\tau] = 0$, since $V_{31}(t), W_{24}(t)$ are non-random and $w_\tau$ is zero mean and independent of $X$. Hence, $\nabla_{W_{24}} L$ is the only non-zero block.

### A.1.2 Proof of Lemma A.2

$$\frac{1}{n^2}\mathbb{E}\left[XX^\top AXX^\top\right] = \frac{1}{n^2}\sum_{i,j}\mathbb{E}\left[x_i x_i^\top A x_j x_j^\top\right]$$

There are $n(n-1)$ terms where $i \neq j$ and $n$ terms with $i = j$. Since $x_i$ and $x_j$ are i.i.d., let $x$ denote either of them. Thus,

$$\frac{1}{n^2}\mathbb{E}\left[XX^\top AXX^\top\right] = \frac{1}{n^2}\left(n(n-1)\mathbb{E}[xx^\top]A\mathbb{E}[xx^\top] + n\,\mathbb{E}\left[xx^\top Axx^\top\right]\right)$$

The first term is the second moments. For the second term we use Isserlis's theorem, by which we have

$$\left(\mathbb{E}\left[xx^\top Axx^\top\right]\right)_{ij} = \sum_{k,l}\mathbb{E}[x_i x_k A_{kl} x_l x_j]$$

$$= \sum_{k,l} A_{kl}\left(\mathbb{E}[x_i x_k]\,\mathbb{E}[x_l x_j] + \mathbb{E}[x_i x_l]\,\mathbb{E}[x_k x_j] + \mathbb{E}[x_i x_j]\,\mathbb{E}[x_l x_k]\right)$$

Assuming $\mathbb{E}[xx^\top] = \Lambda$, we get

$$\mathbb{E}[xx^\top Axx^\top] = \Lambda(A + A^\top)\Lambda + \Lambda\,\mathrm{Tr}(A\Lambda)\,.$$

Therefore we obtain

$$\frac{1}{n^2}\mathbb{E}\left[XX^\top AXX^\top\right] = \frac{n-1}{n^2}\Lambda A\Lambda + \frac{1}{n}\left(\Lambda(A + A^\top)\Lambda + \Lambda\,\mathrm{tr}(A\Lambda)\right)\,.$$

### A.2 Proof of Proposition 3.2

Recall $V_*$ and $W_*$ given by (3.6), as the estimated blocks of the transformer after training. We next rewrite the updates for $w_i$ is a more explicit form:

$$f_{\mathrm{LSA}}(E_i, \theta^*)_{[:,-1]} = \begin{bmatrix} 0_d \\ 0 \\ w_i \\ 1 \end{bmatrix} + VE_i \cdot \frac{E_i^\top W E_{i[:,-1]}}{m}$$

$$= \begin{bmatrix} 0_d \\ 0 \\ w_i \\ 1 \end{bmatrix} + \frac{1}{m}VE_iE_i^\top \begin{bmatrix} cw_i \\ -c \\ 0 \\ 0 \end{bmatrix}$$

$$= \begin{bmatrix} 0_d \\ 0 \\ w_i \\ 1 \end{bmatrix} + \frac{1}{m}V \begin{bmatrix} XX^\top & Xy^\top \\ yX^\top & yy^\top \\ 0_{d\times d} & 0_{m\times m} \\ 0 & 0 \end{bmatrix} \begin{bmatrix} cw_i \\ -c \\ 0 \\ 0 \end{bmatrix}$$

$$= \begin{bmatrix} 0_d \\ 0 \\ w_i \\ 1 \end{bmatrix} + \frac{1}{m} \begin{bmatrix} 0 & 0 \\ 0 & 0 \\ V_{31}XX^\top & V_{31}Xy^\top \\ 0 & 0 \end{bmatrix} \begin{bmatrix} cw_i \\ -c \\ 0 \\ 0 \end{bmatrix}$$

Recalling that $V_{31} = \Gamma^{-1}/c$ we obtain

$$w_{i+1} = w_i - \frac{1}{m}\Gamma^{-1}X_{\mathrm{test}}X_{\mathrm{test}}^\top w_i - \frac{1}{m}\Gamma^{-1}X_{\mathrm{test}}y_{\mathrm{test}}^\top$$

$$= w_i - \frac{1}{m}\Gamma^{-1}X_{\mathrm{test}}X_{\mathrm{test}}^\top(w_i - w_{\mathrm{test}})\,.$$

Rearranging the terms, $w_{i+1} - w_{\mathrm{test}} = (I - \frac{1}{m}\Gamma^{-1}X_{\mathrm{test}}X_{\mathrm{test}}^\top)(w_i - w_{\mathrm{test}})$ which results in

$$w_{k+1} = w_{\mathrm{test}} + (I - \frac{1}{m}\Gamma^{-1}X_{\mathrm{test}}X_{\mathrm{test}}^\top)^k(w_0 - w_{\mathrm{test}})$$

$$= (I - (I - \frac{1}{m}\Gamma^{-1}X_{\mathrm{test}}X_{\mathrm{test}}^\top)^k)w_{\mathrm{test}}\,, \tag{A.9}$$

which completes the proof.

### A.3 PROOF OF THEOREM 3.3

Define the shorthand $\hat{\Lambda} = X_{\text{test}} X_{\text{test}}^\top / m$. We have

$$\mathbb{E}(\|\hat{w} - w_{\text{test}}\|_{\ell_2}^2) = \mathbb{E}(\left\|(I - \Gamma^{-1}\hat{\Lambda})w_{\text{test}}\right\|_{\ell_2}^2) = w_{\text{test}}^\top \mathbb{E}((I - \hat{\Lambda}\Gamma^{-1})(I - \Gamma^{-1}\hat{\Lambda}))w_{\text{test}}$$
$$= w_{\text{test}}^\top (I - \Gamma^{-1}\Lambda - \Lambda\Gamma^{-1} + \mathbb{E}(\hat{\Lambda}\Gamma^{-2}\hat{\Lambda}))w_{\text{test}}$$

Using Lemma A.2 we have

$$\mathbb{E}(\hat{\Lambda}\Gamma^{-2}\hat{\Lambda}) = \frac{m-1}{m}\Lambda\Gamma^{-2}\Lambda + \frac{1}{m}(2\Lambda\Gamma^{-2}\Lambda + \text{tr}(\Lambda\Gamma^{-2})\Lambda)$$
$$= \frac{m+1}{m}\Lambda\Gamma^{-2}\Lambda + \frac{1}{m}\text{tr}(\Lambda\Gamma^{-2})\Lambda \,.$$

Using that $\Gamma^{-1}$ and $\Lambda$ commute and both are symmetric we obtain

$$\mathbb{E}(\|\hat{w} - w_{\text{test}}\|_{\ell_2}^2) = w_{\text{test}}^\top (I - 2\Gamma^{-1}\Lambda + \frac{m+1}{m}\Gamma^{-2}\Lambda^2 + \frac{1}{m}\text{tr}(\Lambda\Gamma^{-2})\Lambda)w_{\text{test}}$$
$$= w_{\text{test}}^\top (I - \Gamma^{-1}\Lambda)^2 w_{\text{test}} + \frac{1}{m}w_{\text{test}}^\top (\Gamma^{-2}\Lambda^2 + \text{tr}(\Lambda\Gamma^{-2})\Lambda)w_{\text{test}}$$

Using the definition $\Gamma = (1 + \frac{1}{n})\Lambda + \frac{1}{n}\text{tr}(\Lambda)I$, it is easy to see that

$$0 \preceq I - \Gamma^{-1}\Lambda \preceq \frac{1}{n}(I + \text{tr}(\Lambda)\Lambda^{-1})$$

Also since $\Gamma^{-1} \preceq \Lambda^{-1}$, we have

$$\Gamma^{-2}\Lambda^2 + \text{tr}(\Lambda\Gamma^{-2})\Lambda \preceq I + \text{tr}(\Lambda^{-1})\Lambda$$

Combining the last two equations, we have

$$\mathbb{E}_{X_{\text{test}}}(\|\hat{w} - w_{\text{test}}\|_{\ell_2}^2) \le w_{\text{test}}^\top \left(\frac{1}{n^2}(I + \text{tr}(\Lambda)\Lambda^{-1})^2 + \frac{1}{m}(I + \text{tr}(\Lambda^{-1})\Lambda)\right)w_{\text{test}}$$

Taking another expectation with respect to $w_{\text{test}} \sim \mathsf{N}(0, I)$ we get

$$\mathbb{E}(\|\hat{w} - w_{\text{test}}\|_{\ell_2}^2) \le \frac{1}{n^2}(d + \text{tr}(\Lambda)^2\text{tr}(\Lambda^{-2}) + 2\text{tr}(\Lambda)\text{tr}(\Lambda^{-1}))) + \frac{1}{m}(d + \text{tr}(\Lambda^{-1})\text{tr}(\Lambda))$$

The claim follows by noting

$$\text{tr}(\Lambda)\text{tr}(\Lambda^{-1}) \le d\frac{\text{tr}(\Lambda)}{\lambda_{\min}},$$
$$\text{tr}(\Lambda)^2\text{tr}(\Lambda^{-2}) \le d\Big(\frac{\text{tr}(\Lambda)}{\lambda_{\min}}\Big)^2,$$

where $\lambda_{\min}$ is the minimum eigenvalue of $\Lambda$.

### A.4 PROOF OF THEOREM 3.4

We define $\hat{\Lambda} := \frac{1}{m}\sum_{i=1}^k x_i x_i^\top$. After $k$ steps generation, we have $w_{k+1} = (I - (I - \Gamma^{-1}\hat{\Lambda})^k)w_{\text{test}}$ and so

$$\mathbb{E}(\|w_{k+1} - w_{\text{test}}\|_{\ell_2}^2) = \mathbb{E}(\left\|(I - \Gamma^{-1}\hat{\Lambda})^k w_{\text{test}}\right\|_{\ell_2}^2) = \mathbb{E}\,\text{tr}((I - \hat{\Lambda}\Gamma^{-1})^k(I - \Gamma^{-1}\hat{\Lambda})^k). \quad \text{(A.10)}$$

Note that $\Gamma^{-1}$ and $\Lambda$ commute and both are symmetric. Therefore $\Gamma^{-1}\Lambda$ is also symmetric. We denote by $\sigma_i$ the eigenvalues of $I - \Gamma^{-1}\Lambda$. The matrix $I - \Gamma^{-1}\hat{\Lambda}$ however is not symmetric. We denote by $\hat{\sigma}_i$ the eigenvalues of $I - \Gamma^{-1}\hat{\Lambda}$. By Weyl's inequality, we have $|\sigma_i - \hat{\sigma}_i| \le \left\|\Gamma^{-1}(\Lambda - \hat{\Lambda})\right\|_{\text{op}} := \delta$.

We then have

$$
\begin{aligned}
\hat{\sigma_i}^{2k} &\leq (\sigma_i + \delta)^{2k} \\
&= \sigma_i^{2k} \left(1 + \frac{\delta}{\sigma_i}\right)^{2k} \\
&= \sigma_i^{2k} \left(1 + \sum_{j=1}^{2k} \binom{2k}{j} \left(\frac{\delta}{\sigma_i}\right)^j\right) \\
&\leq \sigma_i^{2k} \left(1 + \sum_{j=1}^{2k} (2k)^j \left(\frac{\delta}{\sigma_i}\right)^j\right)
\end{aligned}
\tag{A.11}
$$

Define $\tilde{\Delta}_i := \sum_{j=1}^{2k} (\frac{2k\delta}{\sigma_i})^j$. We next proceed by bounding $\mathbb{E}(\tilde{\Delta}_i)$. Observe that $\hat{\Lambda} = \frac{1}{m}\sum_{i\in[m]} x_i x_i^\top$, with $x_i \sim \mathsf{N}(0,\Lambda)$. Using concentration bounds on random matrices with independent sub-gaussian rows (See e.g. (Vershynin, 2010, Eq. 5.26)), we get that with probability at least $1 - 2e^{-ct^2}$,

$$
\left\|\Lambda - \hat{\Lambda}\right\|_{\text{op}} \leq \max(\varepsilon, \varepsilon^2) \left\|\Lambda\right\|_{\text{op}}, \quad \varepsilon = C\sqrt{\frac{d}{m}} + \frac{t}{\sqrt{m}}.
$$

Since $\Lambda$ and so $\Gamma$ have bounded eigenvalues, by adjusting the constants $c, C$ (absorbing $\|\Lambda\|_{\text{op}}$ and $\|\Gamma^{-1}\|_{\text{op}}$ into these constants), we also have that with probability at least $1 - 2e^{-ct^2}$,

$$
\delta := \left\|\Gamma^{-1}(\Lambda - \hat{\Lambda})\right\|_{\text{op}} \leq \max(\varepsilon, \varepsilon^2), \quad \varepsilon = C\sqrt{\frac{d}{m}} + \frac{t}{\sqrt{m}}.
\tag{A.12}
$$

We define the probabilistic event $\mathcal{E} := \{\tilde{\Delta}_i \leq Ck^2\sqrt{d/m}\}$. Obviously, $\mathbb{E}(\tilde{\Delta}_i \mathbf{1}_{\mathcal{E}}) \leq Ck^2\sqrt{d/m}$. We also have $\mathbb{E}(\tilde{\Delta}_i \mathbf{1}_{\mathcal{E}^c}) = \int_{Ck^2\sqrt{d/m}}^{\infty} \mathbb{P}(\tilde{\Delta}_i \geq s)\mathrm{d}s$. Note that by definition of $\tilde{\Delta}_i$, we have

$$
\tilde{\Delta}_i = \sum_{j=1}^{2k} \left(\frac{2k\delta}{\sigma_i}\right)^j \leq 2k\max\left(2k\delta/\sigma_i, (2k\delta/\sigma_i)^{2k}\right) \leq C'k\max(k\delta, (k\delta)^{2k}).
$$

Note that since eigenvalues of $\Lambda$ are upper and lower bounded by constants, so are $\sigma_i$'s. Therefore, we can work with one constant $C'$ that works for all $i \in [d]$.

By virtue of the above bound, if $\tilde{\Delta}_i \geq s$ we have $\delta \geq \min(\frac{s}{k^2}, (\frac{s}{k^{2k+1}})^{\frac{1}{2k}}) \geq \frac{1}{k^2}\min(s, s^{\frac{1}{2k}})$. We next choose $t$ such that for $\varepsilon = C\sqrt{\frac{d}{m}} + \frac{t}{\sqrt{m}}$ we have $\max(\varepsilon, \varepsilon^2) \leq \frac{1}{k^2}\min(s, s^{\frac{1}{2k}})$, so that we can apply the tail bound (A.12).

In addition, for $s \geq Ck^2\sqrt{d/m}$ we have $\frac{1}{k^2}\min(s, s^{\frac{1}{2k}}) \geq C\sqrt{d/m}$, and so it suffices to have $\max(\frac{t}{\sqrt{m}}, \frac{t^2}{m}) \leq \frac{1}{k^2}\min(s, s^{\frac{1}{2k}})$. Therefore, we can set $t = \min(\frac{\sqrt{m}}{k^2}s, \frac{\sqrt{m}}{k^2}s^{\frac{1}{2k}}, \frac{\sqrt{m}}{k}\sqrt{s}, \frac{\sqrt{m}}{k}s^{\frac{1}{4k}})$.

$$
\begin{aligned}
\mathbb{E}(\tilde{\Delta}_i \mathbf{1}_{\mathcal{E}^c}) &= \int_{Ck^2\sqrt{d/m}}^{\infty} \mathbb{P}(\tilde{\Delta}_i \geq s)\mathrm{d}s \\
&\leq \int_{Ck^2\sqrt{d/m}}^{\infty} \mathbb{P}\left(\delta \geq \frac{1}{k^2}\min(s, s^{\frac{1}{2k}})\right) \\
&\leq \int_{Ck^2\sqrt{d/m}}^{\infty} 2\exp(-ct^2)\mathrm{d}s
\end{aligned}
$$

By considering each of the four terms in the minimum operator defining $t$, and following algebraic manipulation, it can be seen that the right-hand side above is $O(k^2\sqrt{d/m})$ and hence,

$$
\mathbb{E}(\tilde{\Delta}_i) = \mathbb{E}(\tilde{\Delta}_i \mathbf{1}_{\mathcal{E}}) + \mathbb{E}(\tilde{\Delta}_i \mathbf{1}_{\mathcal{E}^c}) \leq C(k\sqrt{d/m}),
\tag{A.13}
$$

for a constant $C > 0$ and for all $i \in [d]$. Combining (A.10) and (A.14) and the bound $\tilde{\Delta}_i$, we obtain

$$
\mathbb{E}(\|w_{k+1} - w_{\text{test}}\|_{\ell_2}^2) \le \mathbb{E}[\left\|(I - \hat{\Lambda}\Gamma^{-1})^k\right\|_F^2]
$$

$$
\overset{(a)}{\le} \mathbb{E}(\sum_{i=1}^d \hat{\sigma}_i^{2k})
$$

$$
\overset{(b)}{\le} \sum_{i=1}^d \sigma_i^{2k}(1 + \mathbb{E}(\tilde{\Delta}_i))
$$

$$
\overset{(c)}{\le} \sum_{i=1}^d \sigma_i^{2k}(1 + Ck\sqrt{\tfrac{d}{m}})
$$

$$
= \text{tr}((I - \Gamma^{-1}\Lambda)^{2k})(1 + Ck\sqrt{\tfrac{d}{m}}),
$$

where $(b)$ follows from (A.11) and $(c)$ follows from (A.13). In addition, step $(a)$ follows from the following lemma from Horn & Johnson (1994).

**Lemma A.3** *((Horn & Johnson, 1994, Eq.(3.3.39))) Let A be a given d by d matrix and let m be a given positive integer. For all $p > 0$ we have*

$$
\sum_{i=1}^q \sigma_i(A^m)^p \le \sum_{i=1}^q \sigma_i(A)^{mp}, \quad \text{for } q = 1, \ldots, d,
$$

*where for a matrix B, $\sigma_i(B)$ denotes the singular values of B.*

Step $(a)$ follows by using the above lemma for $A = (I - \hat{\Lambda}\Gamma^{-1})^k$, $m = k$, $p = 2$, $q = d$. This completes the proof.

### A.5   PROOF OF COROLLARY 3.5

Recalling the definition of $\Gamma$ given by (3.5), we have

$$
I - \Gamma^{-1}\Lambda = I - \left[(1 + \tfrac{1}{n})\Lambda + \tfrac{1}{n}\text{tr}(\Lambda)I\right]^{-1}\Lambda
$$

$$
= [(n+1)\Lambda + \text{tr}(\Lambda)I]^{-1}(\Lambda + \text{tr}(\Lambda)I)
$$

$$
\preceq \frac{\lambda_{\min} + \text{tr}(\Lambda)}{(n+1)\lambda_{\min} + \text{tr}(\Lambda)}I
$$

$$
= \frac{1 + \mathsf{Hard}(\Lambda)}{n + 1 + \mathsf{Hard}(\Lambda)}I
$$

$$
= \left(1 + \frac{n}{1 + \mathsf{Hard}(\Lambda)}\right)^{-1} I.
$$

Therefore,

$$
\text{tr}((I - \Gamma^{-1}\Lambda)^{2k}) \le d\left(1 + \frac{n}{1 + \mathsf{Hard}(\Lambda)}\right)^{-2k},
$$

which completes the proof by invoking the result of Theorem 3.4.

### A.6   PROOF OF THEOREM 4.1

The proof follows a long the same lines as in Theorem 3.1. Under the multi-task setting, each feature $x$ is now coming from a mixture of normal distributions. The main modification needed in the proof is on statement of Lemma A.2, which is extended as follows.

**Lemma A.4** *Let $X \in \mathbb{R}^{d \times n}$ with columns drawn i.i.d from a Gaussian mixture distribution, with probability $\pi_\ell$ from $\mathsf{N}(0, \Lambda_\ell)$. Then, for any deterministic matrix A we have*

$$
\mathbb{E}\left(\frac{XX^\top}{n} A \frac{XX^\top}{n}\right) = \frac{n-1}{n}(\sum_\ell \Lambda_\ell \pi_\ell)A(\sum_\ell \Lambda_\ell \pi_\ell) + \frac{1}{n}\sum_\ell (\Lambda_\ell(A + A^\top)\Lambda_\ell + \Lambda_\ell \text{tr}(A\Lambda_\ell))\pi_\ell.
$$

Using Lemma A.4, we have

$$S := \mathbb{E}\left(\frac{XX^\top}{n}\frac{XX^\top}{n}\right) = \frac{n-1}{n}(\sum_\ell \Lambda_\ell \pi_\ell)^2 + \frac{1}{n}\sum_\ell (2\Lambda_\ell^2 + \Lambda_\ell \mathrm{tr}(\Lambda_\ell))\pi_\ell \,.$$

Continuing from (A.2), we have

$$L(\theta) = \frac{c^2}{2}\mathrm{tr}\left(\tilde{V}S\tilde{V}^\top\right) + \frac{d}{2} + c\,\mathrm{tr}(\tilde{V}(\sum_\ell \Lambda_\ell \pi_\ell)))$$

Setting the derivative to zero, we obtain

$$\nabla L(\theta) = \frac{c^2}{2}(\tilde{V}(S + S^\top)) + c\sum_\ell \Lambda_\ell \pi_\ell \,.$$

Solving this equation, using that $S$ is symmetric, we have

$$\tilde{V} = -\frac{1}{c}(\sum_\ell \Lambda_\ell \pi_\ell)S^{-1} = \frac{1}{c}\Gamma^{-1}\,,$$

where the last step follows from the definition of $\Gamma$.

## A.7 PROOF OF PROPOSITION 4.2

The proof is similar to the proof of Theorem 3.4. Recall $\widehat{\Sigma} := \frac{1}{m}X_{\text{test}}X_{\text{test}}^\top = \frac{1}{m}\sum_{i=1}^k x_i x_i^\top$, the empirical covariance of the features in the test prompt. A major difference with the proof of Theorem 3.4, here we have to work with $\Gamma^{-1}\widehat{\Sigma}$ and $\Gamma^{-1}\Sigma$, neither of which are symmetric. To relate the trace of their powers to their singular values, we do a symmetrization step. We write

$$\begin{aligned}
&(I - \Gamma^{-1}\widehat{\Sigma})^k \\
&= (I - \Gamma^{-1}\widehat{\Sigma})(I - \Gamma^{-1}\widehat{\Sigma})\dots(I - \Gamma^{-1}\widehat{\Sigma}) \\
&= \Gamma^{-1/2}\Gamma^{1/2}(I - \Gamma^{-1}\widehat{\Sigma})\Gamma^{-1/2}\Gamma^{1/2}(I - \Gamma^{-1}\widehat{\Sigma})\Gamma^{-1/2}\Gamma^{1/2}\dots\Gamma^{-1/2}\Gamma^{1/2}(I - \Gamma^{-1}\widehat{\Sigma}) \\
&= \Gamma^{-1/2}(I - \Gamma^{-1/2}\widehat{\Sigma}\Gamma^{-1/2})(I - \Gamma^{-1/2}\widehat{\Sigma}\Gamma^{-1/2})\dots(I - \Gamma^{-1/2}\widehat{\Sigma}\Gamma^{-1/2})\Gamma^{1/2} \\
&= \Gamma^{-1/2}(I - \Gamma^{-1/2}\widehat{\Sigma}\Gamma^{-1/2})^k\Gamma^{1/2}\,.
\end{aligned}$$

Hence,

$$\begin{aligned}
\mathrm{tr}((I - \widehat{\Sigma}\Gamma^{-1})^k(I - \Gamma^{-1}\widehat{\Sigma})^k) &= \mathrm{tr}(\Gamma^{1/2}(I - \Gamma^{-1/2}\widehat{\Sigma}\Gamma^{-1/2})^k\Gamma^{-1}(I - \Gamma^{-1/2}\widehat{\Sigma}\Gamma^{-1/2})^k\Gamma^{1/2}) \\
&= \mathrm{tr}((I - \Gamma^{-1/2}\widehat{\Sigma}\Gamma^{-1/2})^k\Gamma^{-1}(I - \Gamma^{-1/2}\widehat{\Sigma}\Gamma^{-1/2})^k\Gamma) \\
&\leq \mathrm{tr}((I - \Gamma^{-1/2}\widehat{\Sigma}\Gamma^{-1/2})^k\Gamma^{-1}(I - \Gamma^{-1/2}\widehat{\Sigma}\Gamma^{-1/2})^k)\mathrm{tr}(\Gamma) \\
&= \mathrm{tr}((I - \Gamma^{-1/2}\widehat{\Sigma}\Gamma^{-1/2})^k\Gamma^{-1}(I - \Gamma^{-1/2}\widehat{\Sigma}\Gamma^{-1/2})^k)\mathrm{tr}(\Gamma) \\
&= \mathrm{tr}((I - \Gamma^{-1/2}\widehat{\Sigma}\Gamma^{-1/2})^{2k}\Gamma^{-1})\mathrm{tr}(\Gamma) \\
&\leq \mathrm{tr}((I - \Gamma^{-1/2}\widehat{\Sigma}\Gamma^{-1/2})^{2k})\mathrm{tr}(\Gamma^{-1})\mathrm{tr}(\Gamma)\,. \qquad \text{(A.14)}
\end{aligned}$$

In the equalities above we used the identity $\mathrm{tr}(AB) = \mathrm{tr}(BA)$. The inequalities follows from the fact that for positive semidefinite matrices $A, B$ we have $\mathrm{tr}(AB) \leq \mathrm{tr}(A)\mathrm{tr}(B)$.

We next denote by $\hat{\sigma}_i$ the eigenvalues of $I - \Gamma^{-1/2}\widehat{\Sigma}\Gamma^{-1/2}$, and by $\sigma_i$ the eigenvalues of $I - \Gamma^{-1/2}\Sigma\Gamma^{-1/2}$. Similar to the proof of Theorem 3.4, we have $\hat{\sigma}_i^{2k} \leq \sigma_i^{2k}(1 + \tilde{\Delta}_i)$ with $\mathbb{E}(\tilde{\Delta}_i) \leq Ck\sqrt{d/m}$ for all $i \in [d]$.

By continuing from (A.14), we get

$$\mathbb{E}[\text{tr}((I - \widehat{\Sigma}\Gamma^{-1})^k(I - \Gamma^{-1}\widehat{\Sigma})^k)] \le \text{tr}(\Gamma^{-1})\text{tr}(\Gamma)\,\mathbb{E}[\text{tr}((I - \Gamma^{-1/2}\widehat{\Sigma}\Gamma^{-1/2})^{2k})]$$

$$\le \text{tr}(\Gamma^{-1})\text{tr}(\Gamma)\,\mathbb{E}[\sum_{i=1}^{d}\hat{\sigma}_i^{2k}]$$

$$\le \text{tr}(\Gamma^{-1})\text{tr}(\Gamma)\sum_{i=1}^{d}\sigma_i^{2k}(1 + \mathbb{E}(\tilde{\Delta}_i))$$

$$\le \text{tr}(\Gamma^{-1})\text{tr}(\Gamma)\sum_{i=1}^{d}\sigma_i^{2k}(1 + Ck\sqrt{d/m})$$

$$= \text{tr}(\Gamma^{-1})\text{tr}(\Gamma)\text{tr}((I - \Gamma^{-1/2}\Sigma\Gamma^{-1/2})^{2k})(1 + o(1)),$$

where in the last step we used that $k\sqrt{d/m} = o(1)$ by our assumption. This concludes the proof.

### A.8 PROOF OF PROPOSITION 4.3

The proof follows from the Markov inequality. Define a discrete random variable $X$ which takes values $\sigma_{\min}(\Lambda_\ell)$ with probability $\pi_\ell$, for $\ell \in [T]$. We then have

$$\mathbb{P}(X \le 2(\varepsilon + \sigma_{\min}(\Sigma))) = \sum_{\ell \in [T]}\pi_\ell \mathbf{1}_{(\sigma_{\min}(\Lambda_\ell) \le 2(\varepsilon + \sigma_{\min}(\Sigma)))} = \sum_{\ell \in D}\pi_\ell.$$

In addition,

$$\mathbb{E}[X] = \sum \pi_\ell \sigma_{\min}(\Lambda_\ell) \le \sigma_{\min}(\sum \pi_\ell \Lambda_\ell) = \sigma_{\min}(\tilde{\Gamma}),$$

by using the convexity of minimum eigenvalue and Jensen's inequality.

Recalling $\Gamma$ from (4.1), we have

$$\Gamma \succeq \frac{n-1}{n}\sum_{\ell \in [T]}\Lambda_\ell \pi_\ell = \frac{n-1}{n}\tilde{\Gamma} \succeq \frac{1}{2}\tilde{\Gamma},$$

for $n \ge 2$. Combining the above two equations, we obtain

$$\mathbb{E}[X] \le 2\sigma_{\min}(\Gamma) \le 2(\sigma_{\min}(\Sigma) + \varepsilon).$$

Therefore,

$$\sum_{\ell \in D}\pi_\ell = \mathbb{P}(X \le 4(\varepsilon + \sigma_{\min}(\Sigma)))$$

$$= 1 - \mathbb{P}(X > 4(\varepsilon + \sigma_{\min}(\Sigma)))$$

$$\ge 1 - \frac{\mathbb{E}[X]}{4(\varepsilon + \sigma_{\min}(\Sigma))}$$

$$\ge 1 - \frac{1}{2} = \frac{1}{2},$$

where we used Markov's inequality in the third step.

## B ADDITIONAL EXPERIMENTS

We report additional experiments on a transformer with a single linear self-attention, when starting training from random initialization and performing CoT with length $k$ during training. Similar to the main paper, the data distribution follows our in-context weight prediction task in Sec. 3.1, where $x_{\tau,i} \sim \mathsf{N}(0, \Lambda)$, $w_\tau \sim \mathsf{N}(0, I_d)$. We choose the token dimensions $d = 10$. During inference, we let model to output $k$ steps before outputting the final predicted weight vector. At each step $i$ we concatenate the embedding with $[0_d, \hat{w}_i, 1]$ as in Eq. 3.1 and input the concatenated embedding matrix to the model. The predicted $w_k$ will be outputted after $k$ steps of CoT.

Fig. 4a, 4b show the test loss during training for $k = 2, 4$. For each $k$, we train the model with $n = 20, 40, 80$. The training and test data are generated from $x_{\tau,i} \sim \mathsf{N}(0, I_d)$, $w_\tau \sim \mathsf{N}(0, I_d)$. We see that for a fixed value of $k$, larger $n$ yields a lower test error, which confirms our theoretical results.

Fig. 4c shows the test loss during training when training distribution is skewed and some directions of the downstream task are not represented enough in the training data. from $\mathsf{N}(0, \Lambda)$ where $\Lambda$ is a skewed covariance matrix with eigenbasis chosen uniformly at random and $i$th eigenvalue proportional to $1/i$. For test, we sample prompt inputs from $\mathsf{N}(0, I_d)$. We use $n = 20$. We see that larger $k$ yields a higher test error. Thus, larger test-time compute hurts the performance when some directions of the downstream data are not enough presented during training.

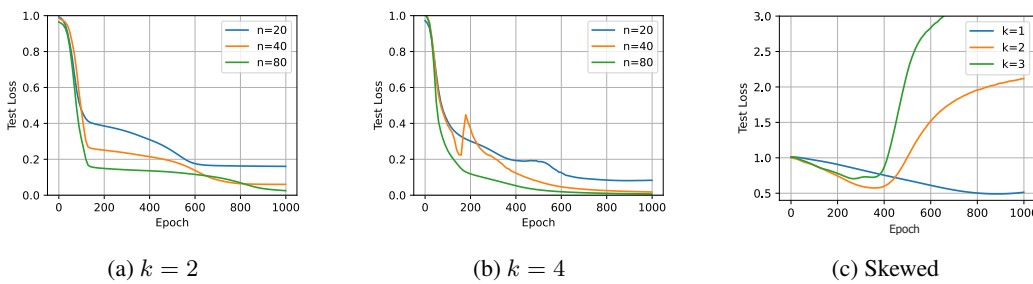

(a) $k = 2$          (b) $k = 4$          (c) Skewed

Figure 4: Transformer with a single linear self-attention. (a), (b) Fixing the test error, by increasing $k$, we can decrease the length of prompts $n$ during training. (c) When some directions of test are not enough represented in training data, more test-time compute hurst the performance.

### B.1   EFFECT OF TASK SELECTION ON TEST TIME SCALING

To demonstrate the improvement we get from our task selection procedure, we consider the set up of Section 5.1, where we generate $w_{\text{test}}$ with i.i.d entries from $N(0, 1)$. During the test time we initialize with $w_0 = 0$ and let the model generate the final estimate of $w_{\text{test}}$ after $k$ step generation. We set the prompt length during training to $n = 50$ and prompt length during the test to $m = 500$. In the plot below we show how the error $\|w_{\text{test}} - w_k\|$ behave for the following task selection procedures: 1) Optimal task selection: We set the probabilities $\pi_\ell$ by solving the optimization problem (4.5). 2) Uniform selection: We select tasks during training with equal probability. 3) Easy task selection: We select only the easy tasks (Easy-Short or Easy-Long as described in Section 5.1) with equal probabilities. In Figure 5, the estimation error for different task selection procedure is plotted versus $k$, the length of generations during test time before outputting the final estimate.

As we see under the optimal choice the error goes down with $k$, while under under the two other procedures the error goes up with $k$, indicating that a proper choice of tasks during training can avoid overthinking, which can occur under other choices of tasks during training.

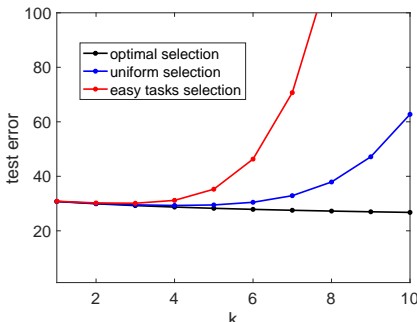

Figure 5: Effect of task selection during training on test-time scaling

