# OpenReview forum: "Understanding the Role of Training Data in Test-Time Scaling"
_ICLR.cc/2026/Conference — ICLR 2026 Poster_

### Official Review · Reviewer_suZT · 2025-10-31

**Soundness:** 2
**Presentation:** 3
**Contribution:** 2
**Rating:** 4
**Confidence:** 3

**Summary:**

This paper studies the relationship between test-time scaling and training data in the context of in-context weight prediction for linear regression. The authors develop a theoretical analysis based on linear self-attention (LSA), derive convergence properties, propose a definition of task hardness, and formulate an optimization framework for task selection. Experiments are conducted with both LSA and GPT-2 architectures to validate the theoretical trends.

**Strengths:**

1. Provides a systematic theoretical analysis with clear derivations of convergence and error bounds.
2. Introduces a task hardness measure based on the covariance matrix, which is simple and interpretable.
3. Offers a formal perspective on task selection, framed as an optimization problem.
4. Addresses the timely topic of test-time scaling, which has attracted significant attention in the community.

**Weaknesses:**

1. Overly idealized setting: The core analysis relies on linear regression and LSA, which are far from realistic large-scale model training.
2. Limited GPT-2 validation: Although GPT-2 experiments are included, they remain confined to the same synthetic weight prediction task, limiting external validity.
3. Simplistic hardness definition: The hardness measure depends only on the smallest eigenvalue, reflecting the limitations of the simplified setting and failing to capture the complexity of real tasks.
4. Insufficient experimental support: The experiments are small-scale and restricted to synthetic tasks, without evidence from realistic reasoning benchmarks.

**Questions:**

See weaknesses.

---

> ### Author Response · Authors · 2025-11-23
>
> Thanks for your review and feedback! We respond to your concerns and questions below.
>
> **Response to Weaknesses**
>
> **Idealized setting:** We agree that our theoretical framework (LSA and linear regression task) is far from the nowaday complex models. However, note that our primary goal in this paper has been on developing a rigorous theoretical understanding of the observations that we have for those complex models, which also gives insight on several important questions discussed in the introduction (in particular if longer CoT/ more test-time compute always help, interaction between test time and train time compute, and the role of training data in avoiding overthinking). Throughout the paper, we have tried to always connect the specifics of the analyzed model to general notions to illustrate how the insights carryover to more complex setting (see bottom of page 5 and top of page 6, as well as our discussion in Section 4.2 in terms of diversity, relevance and hardness of tasks, notions which are not specific to but stems from our analysis of the LSA model).
>
> **Notion of hardness:** We would like to discuss two points here:
>
> **(a)** The hardness measure based on minimum eigenvalue suggests that in-context examples with a more ill-conditioned covariance matrix will be harder. This makes sense as it corresponds to cases where the examples have small projections on some specific directions (those with small eigenvalues) and so cannot easily learn the model component along those directions. This theoretical challenge directly translates to practical learning constraints: the small-eigenvalue directions represent **underrepresented skills or topics** in the pre-training data.
>
> In addition, please note that this measure was based on the bound (3.10), obtained by **some simplification** from the bound before it. Following the proof (line 897-899), one can also define the hardness in terms of $tr(\Lambda) tr(\Lambda^{-1})$. This will be a more robust indicator as it depends on the entire spectrum of the covariance, but it will be more difficult to interpret than the current notion.
>
>
> **(b)** To see how this notion can be applied to complex reasoning tasks, note that based on part (a) of our answer, an individual step within a Chain-of-Thought (CoT) sequence requiring a skill insufficiently represented in the training data correlates directly with a difficult task. Consequently, longer CoTs inherently increase the chance of encountering such an underrepresented skill, and so typically correspond to more challenging tasks.
>
> Crucially, in practice, low performance from the base model on a given task can be an indicator that the task's required skills are underrepresented in the training data and so overthinking would hurt the performance.
>
> **Limitations of Experiments:** We conducted new experiments (added to Appendix B.2 in the revised pdf) where we train Qwen 2.5-7B-Instruct on the OMEGA dataset (Sun et al, 2025). We chose two tasks from the OMEGA dataset, namely GCD and polynomial root reasoning. These tasks are designed such that training on one does not benefit the performance on the other. We fine-tuned the base model (Qwen-Base) with RL separately on the training data of GCD and Poly. We call these models Qwen-GCD and Qwen-Poly. We evaluate both models on the test data of GCD. As expected, for the harder tasks that require longer reasoning, all models have a lower performance. However, we see that while shorter test-time thinking (CoT length less than 1k characters) yields a much better performance (+44.69%) on GCD for Qwen-GCD compared to Qwen-Base, it yields a slightly lower performance on GCD for Qwen-Poly, compared to Qwen-Base (-1.39%). Interestingly, when models reason for longer at test-time (between 1k and 2k characters), Qwen-Poly has a much lower performance (-6.37%) compared to Qwen-Base, while Qwen-GCD outperforms Qwen-Base by 11.2%. **This confirms our theoretical results that when training and test data are aligned, more thinking helps. But, insufficient task coverage in training data makes longer test-time compute harmful.**
>
> **Table:** Average accuracy on GCD for Qwen2.5-7B Instruct (Base), Base model fine-tuned on CGD (Qwen-GCD) and Base model fine-tuned on Poly (Qwen-Poly). For all the models, the accuracy on examples that require longer CoT is lower. This confirms that examples that require longer CoT are generally more difficult. The % in () shows the fraction of test data with the corresponding test-time CoT length. Notably, shorter CoTs (0-1K) considerably improves the performance of Qwen-GCD (75% vs 30.39%) and slightly harms the performance of Qwen-Poly (29% vs 30.39%). Longer CoTs improve the performance of Qwen-GCD (38.4% vs 27.2%) and significantly harm the performance of Qwen-Poly (20.83% vs 27.2%).
>
> |CoT length| 0-1k |1k-2k|
> |---|---|---|
> |Qwen-Base| 30.39% (30% data)| 27.2% (70% data)|
> |Qwen-GCD| 75% (15% data)| 38.4% (85% data)|
> |Qwen-Poly| 29% (32% data)| 20.83% (68% data)|

---

### Official Review · Reviewer_NuKS · 2025-11-01

**Soundness:** 4
**Presentation:** 4
**Contribution:** 3
**Rating:** 8
**Confidence:** 4

**Summary:**

This paper proposes a theoretical lens for studying how training data influences test-time scaling. They propose a simple linear regression prediction task in which the model repeatedly predicts the weight vector given X,y pairs via chain of thought. They demonstrate that increasing test-time compute reduces the in-context examples requirement during training. They also characterize factors that influence the test-time scaling curve, such as training data hardness and diversity of training data.

**Strengths:**

This is a well-written paper. I find the intuitions on task hardness, task selection, and diversity to be particularly helpful in understanding the provided theorems.
* The definitions of task hardness (as a ratio of the sum of variances to minimum eigenvalue) is particularly interesting and to my knowledge quite novel. As the paper states, "an easy task is one that relies on a few dominant skills... while a hard task draws on many skills, reflected in a long-tailed spectrum". In the context of LLMs, task difficulty is usually defined as length/complexity of a problem. The proposed definition provides a new axis of task complexity that relates feature conditioning and data geometry.
* Although the primary focus of the paper is to understand how training data affects test-time scaling curves, the framework naturally explains when overthinking occurs. This interpretability demonstrates the elegance and generality of the theoretical analysis.

Overall, the paper advances our mechanistic understanding of why test-time reasoning improves model performance, and has important consequences for designing the appropriate training data for language models.

**Weaknesses:**

The analysis is mostly confined to the synthetic linear regression task setup. It is not immediately clear how well these definition of task hardness translate to natural language reasoning tasks.

**Questions:**

The theory predicts that the generalization error decays (roughly) as $\frac{1}{n^{2k}}$ - does this scaling hold even for extremely small $n$? In practice, if the model encounters too few examples during training, there could be an error floor that test-time scaling cannot overcome. Is there a theoretical or empirical lower bound on $n$ below which test-time scaling becomes ineffective or unstable?

---

> ### Author Response · Authors · 2025-11-23
>
> Thanks for your review and feedback! You have raised great  points and we respond to your concerns and questions below.
>
> **Response to weaknesses**
>
> As you also summarized our notion of hardness is in terms of the spectrum of the data covariance matrix. This suggests that in-context examples with a more ill-conditioned covariance matrix will be harder. This makes sense as it corresponds to cases where the examples have small projections on some specific directions (those with small eigenvalues) and so cannot easily learn the model component along those directions. This theoretical challenge directly translates to practical learning constraints: the small-eigenvalue directions represent **underrepresented skills or topics** in the model's pre-training data.
>
> **Extension to natural language reasoning:**  For language reasoning tasks, an individual step within a Chain-of-Thought (CoT) sequence requiring a skill insufficiently represented in the training data correlates directly with a difficult task. Consequently, longer CoT trajectories inherently increase the probability of encountering such an underrepresented skill, and so typically correspond to more challenging overall tasks.
>
> Crucially, in practice, low performance from the base model on a given task can be an indicator that the task's required skills are underrepresented in the training data and so overthinking would hurt the performance.  This is confirmed by our new experiments on math reasoning tasks below (see Appendix B.2 for the details of this new experiment).  We can see that Qwen-Poly performs poorly on GCD with shorter test-time CoT. In this case, longer test-time CoT significantly lowers the performance (compared to the base model).
>
> |CoT length | 0-1k |1k-2k|
> |---|---|---|
> |Qwen-Base| 30.39% (30% data)| 27.2% (70% data)|
> |Qwen-GCD| 75% (15% data)| 38.4% (85% data)|
> |Qwen-Poly| 29% (32% data)| 20.83% (68% data)|
>
>
> **Response to Questions:**
>
> Lower bound on $n$: Please note that $n$ indicates the length of prompts during training. That said, we are studying a regime where we observe infinite independent prompts during training (see Equation (3.4) where $B\to \infty$ and we work with population loss). So even if $n$ is small one still gets infinitely many samples to learn the model parameters.

---

### Official Review · Reviewer_8CQr · 2025-11-01

**Soundness:** 3
**Presentation:** 3
**Contribution:** 2
**Rating:** 6
**Confidence:** 3

**Summary:**

This paper discusses the topic of test-time scaling and chain-of-thought prompting for transformers, examining when and why allocating extra compute at inference improves performance.
Motivated by recent systems (e.g., OpenAI’s o1 and DeepSeek R1), the authors analyze transformers trained for in-context weight prediction on linear regression to clarify the training-data conditions under which long chains of thought emerge and help.
They show theoretically that, for a fixed test error, increasing test-time compute can substitute for longer training prompt context length.
They also find that if the relevant skills are insufficiently represented in the training data, additional test-time compute may harm performance.
The work formalizes task hardness via the smallest eigenvalue of the feature covariance matrix and argues that training on diverse, relevant, and hard tasks yields the best gains from test-time scaling.
The analysis connects chain-of-thought prompting to multi-step (pseudo)-Newton’s method and derives scaling laws governing the interaction among test-time compute, context length, and task diversity.
An optimal task-selection strategy for multi-task training is proposed and validated on linear self-attention models and GPT-2, with experiments extending to large, nonlinear transformer architectures.
The authors note limitations: the theory focuses on linear regression and single-layer linear self-attention, and future work should extend to nonlinear data generation and transformers with nonlinear activations.

**Strengths:**

* A coherent framework linking theory and experiments
The paper examines the effectiveness of test-time scaling (longer Chains of Thought) with a back-and-forth between theory (analytical results and scaling laws) and empirical validation (linear self-attention and GPT-2 / larger nonlinear transformers), which strengthens the credibility of its claims.

* Mechanistic identification: CoT as multi-step optimization (pseudo-Newton’s method)
By mapping the inference process to iterative optimization, the paper makes why CoT works more transparent; this unifies prior empirical observations and informs design choices (e.g., preconditioning and stopping criteria).

* A theoretical trade-off between test-time compute and training-time context length
Under a fixed test error, the paper shows that increasing test-time compute can substitute for a longer training context length, providing a principled basis for allocating budget between training and inference.

* A principled metric of task hardness via the smallest eigenvalue of the feature covariance
The work formalizes “hardness” spectrally, enabling computable diagnostics that connect to scaling laws, dataset construction, and benchmark selection.

* Clear conditions for when overthinking can be harmful
The paper specifies that when the necessary skill directions are underrepresented in the training data, increasing test-time compute can degrade performance, offering actionable guidance for safe deployment and monitoring.

* Scaling laws that provide predictability
By deriving scaling relationships among test-time compute, context length, and task diversity (via the features’ covariance spectrum), the paper moves beyond observation to testable predictions that facilitate replication and extension.

**Weaknesses:**

* Narrow applicability of the theory
I understand that the authors explicitly state the limitations (linear regression; single-layer linear self-attention) and future extensions (nonlinear data generation; transformers with nonlinear activations) in the Conclusion and Limitations sections.
However, the main claims (CoT ≈ pseudo-Newton’s method, scaling laws, etc.) are not guaranteed to carry over to recent advances in LLMs, such as different model architectures, like LLaMA-base, GPT-OSS-base, and MoE-based models.
This is extremely important for the contribution of this paper. It would be better to discuss , as part of the main content, how the proposed method is promising for extension to different model configurations, even if confidence is currently limited.

* Dependence on the task “hardness” metric
In Sec. 3, the paper defines hardness as $\mbox{tr} (\Lambda) / \lambda_{min}(\Lambda)$ for the task covariance $\Lambda$, and claims that hardness is based on the smallest eigenvalue of the feature covariance.
However, it seems that relying on a single spectral indicator may be sensitive to preprocessing or representation choices and may not fully capture linguistic complexity or reasoning modes.

* Detectability of the conditions under which overthinking is harmful
In Sec.5, the paper reports observations in which longer CoT hurts.
While the condition is described, a practical method to detect it in advance (a deployable diagnostic or proxy) is not specified.
If my understanding is correct, this indicates that there is still no way to predict this in advance.


* Generality, scale, and diversity of experiments
The experiments are conducted only on LSA and GPT-2 (12 layers, 8 heads, ~9.5M parameters).
It is insightful, but generalization to current large-scale systems and broader task suites (code, long-form reading, knowledge-intensive tasks) remains limited.

**Questions:**

* It remains difficult to separate gains due to the proposed (pseudo-Newton) mechanism from those due to decoding strategies or exploration effects. Is there anything that the authors can discuss on this point?

* Prompt sensitivity / implementation sensitivity
If small implementation choices can flip conclusions, reproducibility is weakened.
Is there anything that the authors can add information on this point?

---

> ### Author Response · Authors · 2025-11-23
>
> Thanks for your review and feedback! You have raised great  points and we respond to your concerns and questions below.
>
> **Response to weaknesses:**
>
> **Narrow applicability of the theory:** We agree that the interpretation of CoT as pseudo-Newton’s method or the specific form of scaling law for our theoretical analysis are not guaranteed to carry over to complex models (LLaMa-based on MoE-based). But we would like to highlight that the primary goal of our analysis is to develop a rigorous theoretical understanding of the observations that we have for those complex models, which also gives insight on several important questions discussed in the introduction (in particular if longer CoT/ more test-time compute always help, interaction between test time and train time compute, and the role of training data in avoiding overthinking). Throughout the paper, we have tried to always connect the specifics of the analyzed model to general notions to illustrate how the insights carryover to more complex setting (see bottom of page 5 and top of page 6, as well as our discussion in Section 4.2 in terms of diversity, relevance and hardness of tasks, notions which are not specific but stems from our analysis of the LSA model). In addition, our new experiments on math reasoning tasks (see Appendix B.2) confirm the validity of our theoretical results on a reasoning task setup. In particular, when the training and test data are aligned, more thinking helps. But, insufficient task coverage in training data makes longer CoTs harmful.
>
> **Hardness measure:** Note that the hardness measure defined in (3.11) is the ratio of trace to minimum singular values (which can be upper bounded by d \times condition number). This suggests that in-context examples with a more ill-conditioned covariance matrix will be harder. This makes sense as it corresponds to cases where the examples have small projections on some specific directions (those with small eigenvalues) and so cannot easily learn the model component along those directions. This theoretical challenge directly translates to practical learning constraints: the small-eigenvalue directions represent **underrepresented skills or topics** in the model's pre-training data.
>
> Regarding your point about being a single spectrum indicator, please note that this measure was based on the bound (3.10), which was obtained by **some simplification** from the bound before it. If you check the proof (line 897-899), you see that one can also define the hardness in terms of $tr(\Lambda) tr(\Lambda^{-1})$. This will be a more robust indicator as it involves all the eigenvalues but will be more involved to work with in our subsequent derivations and is a bit more difficult to interpret than the current notion of hardness.
>
> **Detectability of overthinking:** The insight we got from an analysis and discussed in Remark 4.1 is that if there are task relevant directions which are underrepresented in the training data, more thinking during test time hurts. In other words, the model has learned all relevant skills/directions sufficiently well so thinking more during test time can provide benefits and make the model better in those directions. But if not, the model can confuse itself by thinking more in direction that it has not enough initial understanding. A practical guideline from this is the following: if the performance of a model on a given downstream task (using short test-time CoT) is poor, it implies that the task was not presented enough in the training data. In this case, longer test-time CoT is harmful. On the other hand, if the model has good performance with shorter test-time CoT, then longer CoT can further boost the performance. This is confirmed by our new experiments on math reasoning tasks below (see Appendix B.2 for the details of this new experiment).  We can see that Qwen-Poly performs poorly on GCD with shorter test-time CoT. In this case, longer test-time CoT significantly lowers the performance (compared to the base model).

---

> ### Author Response · Authors · 2025-11-23
>
> **Response to weaknesses (Cont'd)**
>
> **Evaluation on real reasoning benchmarks (New Experiment):** We conducted new experiments (added to Appendix B.2 in the revised pdf) where we train Qwen 2.5-7B-Instruct on the OMEGA dataset (Sun et al, 2025). We chose two tasks from the OMEGA dataset, namely GCD and polynomial root reasoning. These tasks are designed such that training on one does not benefit the performance on the other. We fine-tuned the base model (Qwen-Base) with RL separately on the training data of GCD and Poly. We call these models Qwen-GCD and Qwen-Poly. We evaluate both models on the test data of GCD. As expected, for the harder tasks that require longer reasoning, all models have a lower performance. However, we see that while shorter test-time thinking (CoT length less than 1k characters) yields a much better performance (+44.69%) on GCD for Qwen-GCD compared to Qwen-Base, it yields a slightly lower performance on GCD for Qwen-Poly, compared to Qwen-Base (-1.39%). Interestingly, when models reason for longer at test-time (between 1k and 2k characters), Qwen-Poly has a much lower performance (-6.37%) compared to Qwen-Base, while Qwen-GCD outperforms Qwen-Base by 11.2%. **This confirms our theoretical results that when training and test data are aligned, more thinking helps. But, insufficient task coverage in training data makes longer test-time compute harmful.**
>
> **Table:** Average accuracy on GCD for Qwen2.5-7B Instruct (Base), Base model fine-tuned on CGD (Qwen-GCD) and Base model fine-tuned on Poly (Qwen-Poly). For all the models, the accuracy on examples that require longer CoT is lower (compare the second column to the first column). This confirms that examples that require longer CoT are generally more difficult. The % in () shows the fraction of test data with the corresponding test-time CoT length. Notably, shorter CoTs (0-1K) considerably improves the performance of Qwen-GCD (75% versus 30.39%) and slightly harms the performance of Qwen-Poly (29% versus 30.39%). Longer CoTs improve the performance of Qwen-GCD (38.4% versus 27.2%) and significantly harm the performance of Qwen-Poly (20.83% versus 27.2%).
>
> |CoT length| 0-1k |1k-2k|
> |---|---|---|
> |Qwen-Base| 30.39% (30% data)| 27.2% (70% data)|
> |Qwen-GCD| 75% (15% data)| 38.4% (85% data)|
> |Qwen-Poly| 29% (32% data)| 20.83% (68% data)|
>
> [*]Sun, Yiyou, et al. "OMEGA: Can LLMs Reason Outside the Box in Math? Evaluating Exploratory, Compositional, and Transformative Generalization." arXiv preprint arXiv:2506.18880 (2025).
>
> **Response to Questions**
>
> **Q1.** Our GPT-2 experiments only reflect the gain from the (pseudo-Newton) mechanism. For our GPT-2 experiments, we append zeros to $y_{\tau,i}$ to make it d-dimensional and map the d-dimensional model inputs, i.e., $x_{\tau,i}, y_{\tau,i}$ into the latent embedding space of the Transformer (with 256 dimensions) through a (learnable) linear transformation. Similarly, we map the model output, i.e., $w_{\tau}$ from the latent embedding space of the Transformer to a d-dimensional vector through another (learnable) linear transformation. As we do not use any decoding mechanisms or temperature to sample tokens, our results are not affected by decoding strategies or exploration effects. We clarified this in our revised version in Section 5.
>
> **Q2.** For GPT2 experiments, we trained the model from random initialization with a default learning rate of $10^{-4}$. We didn’t use any implementation tricks, other than training with a curriculum to speed up learning, as is done in (Garg et al, 2023). That is, we trained on a simpler distribution of functions in the beginning (linear functions with weight vectors restricted to a low-dimensional subspace) and gradually increased the function complexity. For the prompts, we used a skewed covariance with eigenvalues $1/i$ for training and sampled our test data from $N(0,I_d)$. The model is trained on fresh prompts each step, thus encountering 1.6M distinct functions and prompts (25k training steps with 64 prompts/batch). We reported the average of 10 runs. Thus, we do expect our results to be reproducible.

---

> > ### Comment · Reviewer_8CQr · 2025-11-27
> >
> > Thank you for providing new experimental results, and adding them in Appendix B.2.
> > I believe that the results of the Qwen 2.5-7B-Instruct model strengthen the claims of this paper. It may be good to add these experimental results in the main text, not just add in the appendix ( but of course, this is not a strong opinion/suggestion.)
> >
> > The rebuttals are mostly convincing, and there are no additional critical concerns from my perspective.
> >
> > One note is that, to be honest, I have not been able to review in detail the correctness and validity of the theorems presented in this paper, and I may have overlooked critical flaws in the theoretical insights.
> > However, I believe the content discussed in this paper is valuable for future LLM research on reasoning (test-time scaling).
> >
> > I will leave my rating unchanged, as my initial rating is already positive for acceptance at the conference.
> >
> >
> >
> > ---
> > Q1: Thank you for the clarification and for adding detailed explanations in the revised version. I now understand that the methods do not rely on any hyperparameters during the decoding phase.
> >
> > Q2: Thank you for the clarification as well. Based on the authors' response, I now understand that the experimental results appear sufficiently robust to be reproducible.
> >
> >
> > Typo: L404-405, "$V31 (0)$" in "... Theorem 3.1 where V 31(0) is set randomly ..." should be "$V_{31} (0)$"

---

### Official Review · Reviewer_cDdC · 2025-11-03

**Soundness:** 3
**Presentation:** 3
**Contribution:** 3
**Rating:** 6
**Confidence:** 3

**Summary:**

The paper studies test‑time scaling in a stylized in‑context learning setting. The core technical model is a one‑layer linear self‑attention transformer trained on an in‑context weight prediction task for linear regression with Gaussian features. They find that CoT induces an iterative update that they interpret as a multi‑step pseudo‑Newton method. They define a task hardness measure, derive error bounds that decay with the CoT depth, and argue that more test‑time compute can compensate for shorter training prompts, and that insufficient skill coverage in training can make longer CoTs harmful.

**Strengths:**

This paper addresses important questions such as whether more inference compute always helps, whether it can trade off against training context length, and what counts as difficult training data.

Proposition 3.2 derives the iterative update which makes the connection between CoT and a preconditioned iterative method precise and inspectable.

The proposed hardness measure is scale‑invariant, emphasizes tail eigen‑mass, and comes with a narrative mapping eigenvectors to skills and eigenvalues to skill strength, which leads to a clear understanding that harder tasks need longer CoT to reach the same error.

**Weaknesses:**

Experiments initialize with the closed‑form optimum using population statistics, so they basically hard‑code the solution instead of learning it, which invalidates empirical support for the learning‑dynamics claims.

All experiments are on synthetic linear tasks, there is no comparison to closed‑form ridge/OLS, no ablation on preconditioner estimation error, no evaluation on real reasoning benchmarks, and no demonstration that the proposed task‑selection improves anything beyond plotting selection weights.

**Questions:**

What is the test‑prompt length in Figures 2?

What is the sample complexity to estimate a usable covariance from a small validation set, and how robust is the solution to estimation noise?

---

> ### Author Response · Authors · 2025-11-23
>
> Thanks for your review and feedback! You have raised great points and we respond to your concerns and questions below.
>
> **Response to Weaknesses:**
>
>  **Hard-coding the solution:** We greatly appreciate it that you brought this into our attention. This is indeed a typo where we wrote in line 414 “we initialize transformer weights as in Eq. (3.6)”.  We initialize it according to Theorem 3.1 ($V(0), W(0)$) where $V_{31}(0)$ is set randomly. We have fixed this in our revised version (uploaded pdf) with blue font.
>
>
>
>  **Comparison to OLS/ridge:** Note that it has been already shown in other works that in-context learning gets comparable performance to optimal least square (see Garg* et. al 2023). That said we can also see it quite straightforward from our analysis. Consider the LSA at global optimum V* and W* (Eq (3.6)). Then during test time if there is no thinking (k=1) (so this is vanilla in-context learning) then by equation (3.8) the estimate is given by $w_1 = \Gamma^{-1}(\frac{1}{m} X_{\rm test} X_{\rm test}^T) w_{\rm test} = \Gamma^{-1} (\frac{1}{m} X_{\rm test} y_{\rm test})$, as we initialize with $w_0 = 0$ and $y_{\rm test} = X_{\rm test} w_{\rm test}$.  The OLS solution is given by $(\frac{1}{m} X_{\rm test} X_{\rm test}^T)^{-1} y_{\rm test}$. So this is similar to OLS where the empirical covariance is estimated by $\Gamma$ (the population covariance of training data).
>
> [*]Garg, Shivam, et al. "What can transformers learn in-context? a case study of simple function classes." Advances in neural information processing systems 35 (2022): 30583-30598.
>
>
>  **Impact of task selection on test time scaling (New Experiment):** To demonstrate the improvement we get from our task selection procedures, we conducted the following experiment. Consider the set up of Section 5.1, where we generate $w_{\rm test}$ with i.i.d entries from $N(0,1)$. During the test time we initialize with $w0= 0$ and let the model generate the final estimate of $w_{\rm test}$ after k step generation. We set the prompt length during training to $n= 50$ and prompt length during the test to $m = 500$.  Please see Appendix B.1 in our revised pdf of the paper for the details of the experiment.
>
> Figure 5 (in Appendix B.1) shows the effect of different task selection procedures on the test time scaling. In the plot below we show how the error $||w_{\rm test} - w_k||$ behave for different task selection probabilities. As we see under the optimal choice (cf. Equation 4.5) the error goes down with $k$, white under uniform probabilities and a procedure which  only chooses easy tasks during training, the error goes up with $k$. This experiment demonstrates the role of our task selection during training on test time scaling. In particular, while overthinking is avoided under our procedure, it occurs under the other two task selection procedures.
>
>  **Evaluation on real reasoning benchmarks (New Experiment):** We conducted new experiments (added to Appendix B.2 in the revised pdf) where we train Qwen 2.5-7B-Instruct on the OMEGA dataset (Sun et al, 2025). We chose two tasks from the OMEGA dataset, namely GCD and polynomial root reasoning. These tasks are designed such that training on one does not benefit the performance on the other. We fine-tuned the base model (Qwen-Base) with RL separately on the training data of GCD and Poly. We call these models Qwen-GCD and Qwen-Poly. We evaluate both models on the test data of GCD. As expected, for the harder tasks that require longer reasoning, all models have a lower performance. However, we see that while shorter test-time thinking (CoT length less than 1k characters) yields a much better performance (+44.69%) on GCD for Qwen-GCD compared to Qwen-Base, it yields a slightly lower performance on GCD for Qwen-Poly, compared to Qwen-Base (-1.39%). Interestingly, when models reason for longer at test-time (between 1k and 2k characters), Qwen-Poly has a much lower performance (-6.37%) compared to Qwen-Base, while Qwen-GCD outperforms Qwen-Base by 11.2%. **This confirms our theoretical results that when training and test data are aligned, more thinking helps. But, insufficient task coverage in training data makes longer test-time compute harmful.**
>
> Please see the next official comment for the table (or see Appendix B.2 in the revised pdf.)

---

> ### Author Response · Authors · 2025-11-23
>
> This is the table also reported in Appendix B.2 of the revised pdf.
>
> **Table:** Average accuracy on GCD for Qwen2.5-7B Instruct (Base), Base model fine-tuned on CGD (Qwen-GCD) and Base model fine-tuned on Poly (Qwen-Poly). For all the models, the accuracy on examples that require longer CoT is lower (compare the second column to the first column). This confirms that examples that require longer CoT are generally more difficult. The % in () shows the fraction of test data with the corresponding test-time CoT length. Notably, shorter CoTs (0-1K) considerably improves the performance of Qwen-GCD (75% versus 30.39%) and slightly harms the performance of Qwen-Poly (29% versus 30.39%). Longer CoTs improve the performance of Qwen-GCD (38.4% versus 27.2%) and significantly harm the performance of Qwen-Poly (20.83% versus 27.2%).
>
> |CoT length| 0-1k |1k-2k|
> |---|---|---|
> |Qwen-Base| 30.39% (30% data)| 27.2% (70% data)|
> |Qwen-GCD| 75% (15% data)| 38.4% (85% data)|
> |Qwen-Poly| 29% (32% data)| 20.83% (68% data)|
>
> [*]Sun, Yiyou, et al. "OMEGA: Can LLMs Reason Outside the Box in Math? Evaluating Exploratory, Compositional, and Transformative Generalization." arXiv preprint arXiv:2506.18880 (2025).
>
> **Response to Questions**
>
> **Q1.** test and train prompt lengths are the same, and are shown by "$n$" in the legend.
>
> **Q2.** The sample complexity depends on the norm in which you want to get a good approximation. For example if you have n iid  vectors $x_i\in R^d$ from a subgaussian distribution with covariance $\Sigma$. Denote the sample covariance $\widehat{\Sigma} = 1/n \sum x_i x_i^T$, then the maximum entrywise deviation is bounded as$| \Sigma - \widehat{\Sigma}|_{\infty} = O(\sqrt{\frac{\log d}{n}})$. But if you want to have a good estimate in operator norm, then you need much larger sample size, $\|\Sigma - \hat{\Sigma}\|{\rm op} = O(\sqrt{d/n})$.  In particular, our assumption $m = \Omega(k^2 d)$ in Theorem 3.4 comes from this type of concentration bound (check Vershynin 2010 also listed in the References).

---

### Meta-Review · Area_Chair_7VzZ · 2026-01-10

**Summary:**

This paper develops a clean theoretical framework for test-time scaling / longer CoT in a stylized but insightful setting: transformers trained for in-context weight prediction for linear regression, where longer CoT corresponds to an iterative update interpretable as a (preconditioned) optimization method. The main takeaways are (i) a principled trade-off between inference compute and training prompt length at fixed error, (ii) a formalization of when more thinking can hurt under insufficient “skill” coverage in training, and (iii) a spectral notion of task hardness (via the feature covariance spectrum) that supports concrete guidance for task selection and diversity. Reviewers generally found the analysis well-written and conceptually useful. Overall, despite remaining limitations in breadth, the paper offers a coherent theoretical lens and testable predictions that are likely to be valuable to the community.

**Reviewer Concerns:**

Major concerns addressed by rebuttal
Potential “hard-coding” of solution through initialization: Authors state this was a typo and clarify that initialization follows the theorem with random components
Lack of evidence beyond synthetic linear tasks: Added Qwen2.5-7B-Instruct experiment on OMEGA (GCD vs polynomial-root training) provides supportive evidence that (a) aligned training benefits from longer CoT while (b) misaligned training can be harmed by longer CoT.

Concerns that remain outstanding
Breadth / generality: Core theory is still specialized. The new Qwen result helps, but the overall experimental scope remains limited relative to the breadth of claims often associated with “test-time scaling” in frontier LLMs.
Hardness metric / practical diagnostics: The spectral hardness measure is well-motivated in the model, but translation to NLP settings remains mostly conceptual.

**Reviewer Scores:**

Reviewer cDdC (6 → ~6/7): The rebuttal resolves the most serious concerns
Reviewer 8CQr (6 → 6): Reviewer explicitly states rating is unchanged
Reviewer NuKS (8 → 8):
Reviewer suZT (4 → ~5): The primary criticisms are partially mitigated by the added Qwen experiment

---

### Decision · Program_Chairs · 2026-01-26

Accept (Poster)